# Analyzing Global Geopolitical Stability in Terms of World Trade Network Analysis

Georgios D. Papadopoulos [1,*], Lykourgos Magafas [1,*], Konstantinos Demertzis [2] and Ioannis Antoniou [3]

1 Laboratory of Complex Systems, Department of Physics, Faculty of Sciences, International Hellenic University, Campus St. Loukas, 654 04 Kavala, Greece

2 School of Science & Technology, Informatics Studies, Hellenic Open University, 263 35 Patra, Greece; demertzis.konstantinos@ac.eap.gr

3 School of Mathematics, Faculty of Physics and Mathematics, Aristotle University of Thessaloniki, 541 24 Thessaloniki, Greece; iantonio@math.auth.gr

* Correspondence: dgepapa@physics.ihu.gr (G.D.P.); magafas@physics.ihu.gr (L.M.); Tel.: +30-6932808425 (G.D.P.)

**Abstract:** The global economy operates as a complex and interconnected system, necessitating the application of sophisticated network methods for analysis. This study examines economic data from all countries across the globe, representing each country as a node and its exports as links, covering the period from 2008 to 2019. Through the computation of relevant indices, we can discern shifts in countries' positions within the world trade network. By interpreting these changes through geopolitical perspectives, we can gain a deeper understanding of their root causes. The analysis reveals a notable trend of slow growth in the world trade network. Additionally, an intriguing observation emerges: countries naturally form stable groups, shedding light on the underlying structure of global trade relations. Furthermore, this research highlights the trade balance as a reflection of geopolitical strength, making it a valuable contribution to the study of the evolution of global geopolitical stability.

**Keywords:** annual countries exports; network analysis; geopolitical stability

## 1. Introduction

Nowadays, the use of new theories and techniques enables us to comprehend our world and the changes that take place in it more effectively [1,2]. For example, the changes to the environment that are caused by human activity appear initially on a local level and then emerge on a global scale. Remote areas of our planet seem to influence each other.

Similar phenomena manifest also in economy in a more intense way, resulting in the shift to globalization. Events of a different nature and scale (political, social, financial, etc.) in a country bring about successive changes in its economic field, which eventually spread to its partners and to the whole world [3].

The global economy is closely linked to global geopolitical stability, and any change in one brings about small or large changes in the other. The ways through which this connection is made are as follows [4]:

1. Economic interdependence: All countries of the world are connected via the network of distribution of goods, raw materials, and services. The development of countries and the improvement of their citizens' standard of living are based on exports and the inflow of capital into them. In times when there is a disruption in the global product supply chain and consequently a slowdown in trade, there are negative effects around the world, such as, e.g., decrease in income, a decrease in jobs, recession, social instability, etc. The consequence of these are the tensions and conflicts between competing countries, with the worst consequence being war;

2. The constant competition for resources: Today, the most frequent reason for the creation of friction and wars between countries is access to natural resources, especially

oil and natural gas. These minerals are used as leverage to apply more favorable regulations and policies to the countries that have them in abundance at the expense of other wronged countries. On the other hand, economic development leads to the overexploitation of natural resources, resulting in the creation of environmental issues that in turn worsen people's quality of life, thus contributing to instability;

3.  Economic inequalities: The concentration of natural resources and consequently the concentration of wealth by a privileged few increases citizens' frustration with the political system and strengthens the feeling of social injustice. This leads to the emergence of extreme tendencies and extremist ideologies;

4.  Financial crises: The global recession that occurs after major financial crises can often be a precursor to a long period of tensions and war conflicts between countries with high levels of inequality and weak institutions (1929 crisis–World War II);

5.  Cooperation between countries: Economically interdependent countries are more likely to coalesce around common positions, thereby influencing decisions concerning the global community, such as, e.g., climate change, global pandemics, or the imposition of measures in cases of authoritarian governments, civil strife, etc.

Consequently, global economy functions as a complex network where the countries constitute its nodes and the commercial transactions among the countries its edges. The analysis of this network has constituted a research field for the scientific community for some years now, attracting the interest of researchers from different fields such as physics, mathematics, information technology, economics, etc. [1,5].

## 2. Research Studies on World Economic Network

The study of the world economic network is based on the complex networks theory as developed in the last 23 years [6].

In 1992, D. Smith and D. White, using financial data from 1965 to 1980 about imports for more than 100 countries, showed that the world economic network is structured in layers (core–periphery) consisting of groups of countries of decreasing order of influence. They found out that there is a tendency in the world economy to expand the core and reduce the number of peripheral countries [7].

M.A. Serrano and M. Boguna in 2003 created a complex directed network using import–export data for the 40 most marketable products among countries for the year 2000 and presented the first empirical characterization of the world-trading web, calculating some typical properties that characterize complex networks such as distribution of grades, clustering coefficient, etc. Their calculations showed that the world trade network cannot be described as a classical random network because it displays the typical properties of a scale-free network with a small-world property and is highly clustered [8].

In 2005, D. Garlaschelli and M.I. Loffredo studied the topological properties of the world trade web (WTW), which are closely related to the gross domestic product (GDP) of world countries from 1950 to 1996 and took into consideration the directional nature of the international trade routes and the time dependence of the parameters describing the topology WTW [9].

Bhattacharya et al. in 2008 investigated the ranges of different quantities relevant to the world economic network (WEN), expanding their data from 1948 to 2000 and assuming that WEN is an undirected network with the commercial flows among countries as its edge weights. They noticed that WEN remained unchanged over a span of 53 years, implying robustness or universality. Among other things, they also showed that a big part of the world trade is controlled by a small club of rich countries that is shrinking as time goes by [10].

Moving on, Fagiolo et al. showed in 2009 that for the years 1981–2000, apart from the structure of core–periphery, weak trade bonds characterize most countries, and rich countries have stronger trade bonds and are more tightly organized in groups. They also noticed that the world trade web (WTW) is statistically more clustered than if it were

random, and all the properties of the network are remarkably stable through the years and not dependent on the weighting procedure [11].

Globalization, which has occurred in the last decades, creates changes not only in terms of economics but also to the structural properties of socio-economic networks towards greater complexity. The increase in rank and connectivity of common economic networks causes greater uncertainty and instability dangers. D. Hossu et al. in 2009 applied network measurements to the world trade data and proposed some effective means of managing the complexity of the world economic network based on efficient models [12].

The economic crisis of 2008 in the USA and its dissemination around the world led the greatest part of the researchers' attempts to search for the countries that influence the global economy the most, that, is the countries that dominate exports. Towards this direction, the investigation of Reyes et al. in 2011 using data of transboundary bank loans for 184 countries for the years 1978–2009 showed that connectivity tends to decrease during and after systemic crises of national debt, using measurements of centrality, connectivity, and grouping. The world economic crisis of 2008–2009 is referred to as an unusually big disruption in the transboundary bank network [13].

However, since the economic crisis of 2008 and onwards, other cases of countries with smaller GDPs and fewer exports were studied as well as how and how much the topology of the world macroeconomic network influences the spread of economic crises. In 2011, Lee et al. noted that the role of an individual country in the dissemination of crises does not only depend on its gross macroeconomic ability but also on its regional and world connectivity profile in the framework of the world economic network. The researchers expressed the opinion that the grouping of weaker countries on a peripheral scale may deteriorate the expansion of crises significantly, but on the other hand, this global network structure shows greater tolerance to extreme crises as compared to a more "globalized" random network [14].

Furthermore, in a study by L. De Benedictis et al. in 2013, an economic dataset for 178 countries from 1995 to 2010 was used, and the topology and properties of the world trade network were investigated. Their study concerned the weighted and unweighted version of the world network and at the same time offered visualized network models for various commercial products [15].

In a study published recently in 2022, D.G. Demiral and M.I. Yenilmez searched for the effects of the COVID-19 pandemic on global trade, by building networks with the 50 largest exporting countries. Degree, closeness, and betweenness centralities were calculated, and they found the existence of four or five clusters around countries with strong economies [16].

The scope of this study is to analyze the global economy as a complex system of interdependent factors by employing complex network methods. It focuses on economic data from all countries worldwide, treating each country as a node and its exports as links. The analysis primarily revolves around identifying shifts in countries' positions within the network and understanding the potential causes behind these changes through a geopolitical lens.

The motivation of our work is to extend the results of previous investigations in the following directions: (1) include the period 2008–2019, (2) characterize more precisely the world trade network by including additional indices (clustering coefficient, average path length, eccentricity, betweenness centrality, modularity, and number of communities), and (3) highlight geopolitical implications. More specifically, we address the following research questions:

Q1.  How does the importance of countries in the global economy manifest in terms of network properties?

Q2.  How do collaborations between countries and global geopolitical stability appear as network properties? Can we identify groups of countries with "stronger" links among them?

Q3. Can changes in the global economy be manifested and observed as changes in network properties? In particular, how does the trend towards globalization in the global economy manifest in terms of network metrics?

Q4. What are the geopolitical implications of the answers to questions Q1–Q3?

The methodology to address the above questions Q1–Q4 is as follows:

1. We employ complex network analysis tools for the study of geopolitical stability, particularly in the context of the global trading system using data on international trade;

2. We evaluate selected local and global network indicators of connectivity as well as average values, and we indicate the meaning of these values in the context of financial data;

3. We identify countries with weak and strong bonds, the related power dynamics, and the resulting geopolitical tensions that emerged in subsequent years.

The network analysis tools are described in Section 3. The results are presented in Section 5 and analyzed in Section 6.

## 3. Methodology of the Research—Materials and Methods

The network indicators used to capture aspects of global economy are presented in order to correlate them with indicators of the world economy (addressing Q1, Q2) and draw conclusions (addressing Q3, Q4). Centralities highlight the importance of some particular nodes in the networks [1,6,17,18].

### 3.1. Network Specification

A network with $N$ nodes is described by the weight matrix with elements $w_{\kappa\lambda}$, $\kappa$, $\lambda$ = 1, 2, ..., $N$. In our case, the nodes are countries/economic entities participating in the world trade. Each weight $w_{\kappa\lambda}$ is the economic (export) flow from node $\kappa$ to node $\lambda$ and takes non-negative values. As there is no economic flow from a country to itself, the diagonal elements are zero $w_{\kappa\kappa} = 0$, $\kappa$ = 1, 2, ..., $N$.

The adjacency matrix $[a_{\kappa\lambda}]$, $\kappa$, $\lambda$ = 1, 2, ..., $N$ indicates the presence of economic flows between nodes:

$$a_{\kappa\lambda} = \begin{cases} 1, \; if \; w_{\kappa\lambda} > 0 \\ 0 \; if \; w_{\kappa\lambda} = 0 \end{cases} \tag{1}$$

The number of links $E = \sum_{\kappa,\lambda=1}^{N} a_{\kappa\lambda} [\![ \kappa \neq \lambda ]\!]$ indicates the number of present economic flows. Here, we use the *Iverson bracket*, a simple, computationally useful notation introduced by Knuth [19] and defined for any statement as follows:

$$[\![ Statement ]\!] = \begin{cases} 1, \; if \; Statement \; is \; True \\ 0, \; if \; Statement \; is \; Not \; True \end{cases} \tag{2}$$

### 3.2. Density

The fraction of the number $E$ of present links to the number $N(N-1)$ of possible links between nodes is the density of the network:

$$\rho = \frac{E}{N(N-1)} \tag{3}$$

The density indicates how dense or sparse the network is.

### 3.3. Degree

The numbers of incoming and outgoing neighbors of a node is known as the in and out degrees of the node.

$$deg_{\kappa}^{in} = \sum_{\lambda=1}^{N} \alpha_{\lambda\kappa} \tag{4}$$

$$deg_\kappa^{out} = \sum_{\lambda=1}^{N} \alpha_{\kappa\lambda} \tag{5}$$

The sum of in and out degrees is the (total) degree of the node:

$$deg_\kappa = deg_\kappa^{in} + deg_\kappa^{out} = \sum_{\lambda=1}^{N} (\alpha_{\kappa\lambda} + \alpha_{\lambda\kappa}) \tag{6}$$

We also consider the average degree and the average weighted degree or average strength of the network:

$$\overline{deg} = \frac{\sum_{\lambda=1}^{N}(\alpha_{\kappa\lambda} + \alpha_{\lambda\kappa})}{N} \tag{7}$$

$$\overline{deg}^{\{w\}} = \frac{\sum_{\lambda=1}^{N}(w_{\kappa\lambda} + w_{\lambda\kappa})}{N} \tag{8}$$

In our case, the out degree of a country indicates the number of export flows to other countries, and the weighted degree indicates the amount of trade flows in USD to other countries. The average weighted degree shows the condition of the world economic trade directly. This is because net amounts of a country's exports are used (in USD) as weights on the edges of the respective node. Thus, every modification in the amount of exports of a country and especially of those dominating the world trade is depicted in this particular indicator in exactly the same way as it happens in the changes of the economic indicator GDP.

The (out) flow index of each node is defined as the difference between out and in degree:

$$deg_\kappa^{\{w\}flow} = deg_\kappa^{\{w\}out} - deg_\kappa^{\{w\}in} \quad \kappa = 1,\ldots,N \tag{9}$$

The superscript $\{w\}$ refers to the (weighted) WTN.

*3.4. Clustering*

The fraction of the number $E_\kappa$ of links between the first neighbors of a node $\kappa$ to the number of possible links among them is the clustering coefficient of node $\kappa$:

$$clu_\kappa = \begin{cases} \frac{E_\kappa}{N_\kappa(N_\kappa-1)}, & if\ N_\kappa \geq 2 \\ 0, & otherwise \end{cases} \tag{10}$$

where

$$E_\kappa = \sum_{\mu=1}^{N}\sum_{\nu=1}^{N} \alpha_{\mu\nu} \cdot [\![\alpha_{\kappa\mu} + \alpha_{\mu\kappa} \geq 1]\!] \cdot [\![\alpha_{\kappa\nu} + \alpha_{\nu\kappa} \geq 1]\!] \cdot [\![\mu \neq \nu]\!] \cdot [\![\mu \neq \kappa]\!] \cdot [\![\nu \neq \kappa]\!] \tag{11}$$

is the number of links between the first neighbors of a node $\kappa$.

$$N_\kappa = \sum_{\nu=1}^{N} [\![\alpha_{\kappa\nu} + \alpha_{\nu\kappa} \geq 1]\!] \cdot [\![\nu \neq \kappa]\!] \tag{12}$$

is the number of nodes adjacent to $\kappa$.

The average clustering coefficient of the network is given:

$$\overline{clu} = \frac{\sum_{\kappa=1}^{N} clu_\kappa}{N} \tag{13}$$

*3.5. Groups within Networks Modularity*

The problem of finding groups (communities, clusters, and modules) of nodes within networks is the clustering problem in the context of graphs [6]. An efficient way to identify

the presence of groups is to characterize its partition of nodes in the network by the value of modularity. For each partition, the value of Modularity, introduced by Newmann, indicates the distinguishability of groups, taking values in the interval [−1, 1] [6]. The connectivity among the nodes within each group is much higher, compared to the connectivity among the nodes in different groups, in networks with high modularity. We compute the modularity for the weighted network. The absence of the weights of the exports is not a proper way to address research questions Q1–Q4 (Section 2) because the mere presence of exports does not correspond to the real WTN. The difference between the WTN and the corresponding unweighted graph will be shown for comparison.

Optimal clusters are those with maximal modularity [6]. We employed the Louvain method for community identification corresponding to maximal modularity. The corresponding algorithm is included in the software GEPHI [20]. In the case of world trade networks, modularity analysis allows to identify groups of highly connected countries as well as which countries change groups more frequently.

### 3.6. Distance and Topology

The topology of network is described by the distance of nodes, the diameter, the eccentricity, and the average path length, defined as follows [6,17,18]:

The directed distance from node $\kappa$ to node $\lambda$ is the length in steps/edges of the shorter directed path from node $\kappa$ to node $\lambda$ within the network.

The diameter of the network is defined as the longest directed distance between any two nodes that exist in the network.

Out eccentricity of a node is the length of the largest directed path from the node to the other nodes of the network. The nodes with the smallest eccentricity define the center of the whole network.

Average path length $\bar{d}$ of the network is the average directed distance between any pair of nodes, indicating the efficiency of information or mass transport on a network.

### 3.7. Closeness Centrality

The out closeness centrality of node $\kappa$ is the inverse average distance from the node $\kappa$ to the other nodes of the network [21].

$$\overline{c\ell o}_{\kappa}^{out} = \frac{1}{\bar{d}_\kappa} \tag{14}$$

where $\bar{d}_\kappa = \frac{\Sigma_{\lambda=1}^{N} d_{\kappa\lambda}}{N}$ is the average directed distance from node $\kappa$ to other nodes.

### 3.8. Betweenness Centrality

The presence of nodes acting as liaisons in the network is estimated in terms of the betweenness centrality of each node [22,23]:

$$B_\nu = \frac{1}{(N-1)\cdot(N-2)} \sum_{\kappa,\lambda=1}^{N} \left( \frac{\sigma_{\kappa(\nu)\lambda}}{\sigma_{\kappa\lambda}} \cdot [\![\kappa \neq \lambda]\!] \cdot [\![\kappa \neq \nu]\!] \cdot [\![\nu \neq \kappa]\!] \right) \tag{15}$$

where $\sigma_{\kappa\lambda}$ is the number of directed paths from node $\kappa$ to node $\lambda$, and $\sigma_{\kappa(\nu)\lambda}$ is the number of directed paths from node $\kappa$ to node $\lambda$ passing through node $\nu$.

The *Iverson brackets* in the sum exclude the possibilities that the node $\nu$ does not coincide with the end nodes $\kappa$, $\lambda$ and that the end nodes $\kappa$, $\lambda$ are distinct.

Networks with high betweenness centrality include many nodes in the paths connecting pairs of nodes in the network. Nodes with high betweenness centrality are acting as liaisons—facilitators—intermediaries between the nodes of the network. In the case of the world exports network, countries with high betweenness are most often intermediaries in the trade links of the other countries and are also involved in the transportation of goods.

### 3.9. Eigenvector Centrality

The out eigenvector centrality $eig_v^{\text{out}}$ of each node $v$ is the $v$-*th* component of the *Perron–Frobenius eigenvector* of the adjacency matrix $[\alpha]$. We remind that the *Perron–Frobenius eigenvector* of a matrix is the normalized eigenvector associated with the dominant eigenvalue (the eigenvalue with the maximum absolute value). Nodes with high eigenvector centrality are considered as regulators of the network, as they are connected [6,17,18].

### 4. Datasets

We used free access data developed by *The World Bank* in collaboration with the *United Nations Conference on Trade and Development* (UNCTAD) and in consultation with organizations such as the *World Trade Center*, the *United Nations Statistics Division* (UNSD), and the *World Trade Organization* (WTO) [24].

The data describe the exports of the member countries of the World Bank from 2008 to 2019 in thousands of dollars (USD) and current values. The destinations of export products include independent countries and other political and/or financial entities that are not independent countries of the UN, such as offshore-occupied territories of Britain and France; sub-national administrative divisions and provinces of the United States, of China (Other Asia Nes/Taiwan), and other countries; as well as protected areas of scientific interest. The total number of these destinations is 243.

In the years 2008–2019, no export economic data were published for various reasons (wars, riots, and imposition of restrictions) either for specific years or for a continuous period for some countries. Hence, in the network planning, these particular countries appear either as nodes with no outcoming links (exports) or as isolated nodes. In the latter case, the isolated nodes were removed from the analysis so that the results of the measurements were not influenced in any way. All these result, on the one hand, in the number of nodes not being stable and, on the other hand, in a fluctuating number of edges linking the nodes.

Moreover, in some cases, there was a shortage of financial data for the more recent years (data are available for 10 countries until 2017 and for 24 countries until 2018). In the case that this lack concerned a country with a considerable volume of exports (for example, for Ukraine, there are no data for 2019), then there was a considerable reduction in the number of the network's links for this particular year. Data were missing for less than five countries up to 2016 (Figure 1). Afterwards, data were not available for more countries.

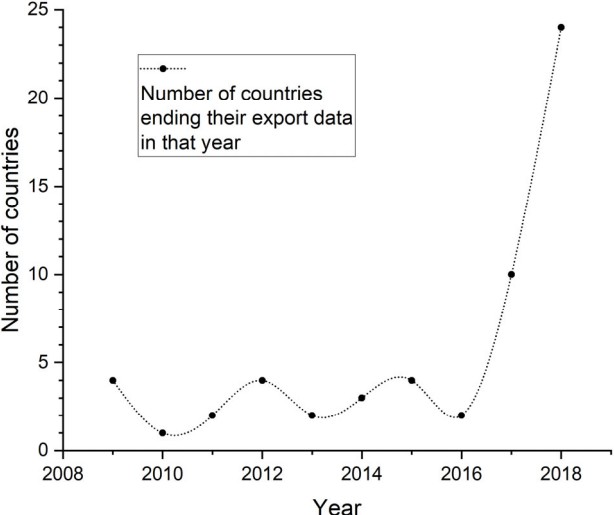

**Figure 1.** The number of countries lacking export data during 2008–2018.

The countries with missing export data are presented in Table 1. It became clear that these are cases with small transactions volume, so their impact on the calculation

of network measures was negligible. However, the presence of a country as a node in the whole network was not influenced by the above-mentioned lack of data, as it is quite possible that there was some financial flow towards this country from another country in the same network.

**Table 1.** The countries with missing export data during 2008–2018.

| | | | | | | | |
|---|---|---|---|---|---|---|---|
| Anguilla | 2008 | Micronesia, Fed. Sts. | 2013 | Lesotho | 2017 | Iran, Islamic Rep. | 2018 |
| Grenada | 2008 | Venezuela | 2013 | Mali | 2017 | Lebanon | 2018 |
| Netherlands Antilles | 2008 | Iraq | 2014 | Nepal | 2017 | Libya | 2018 |
| Djibouti | 2009 | Montserrat | 2014 | Panama | 2017 | Maldives | 2018 |
| Faroe Islands | 2009 | Tonga | 2014 | Sri Lanka | 2017 | Montenegro | 2018 |
| Gabon | 2009 | Bangladesh | 2015 | St. Kitts and Nevis | 2017 | Mozambique | 2018 |
| Mayotte | 2009 | French Polynesia | 2015 | Albania | 2018 | Niger | 2018 |
| Syrian Arab Republic | 2010 | New Caledonia | 2015 | Andorra | 2018 | Oman | 2018 |
| Cook Islands | 2011 | Trinidad and Tobago | 2015 | Angola | 2018 | Palau | 2018 |
| Vanuatu | 2011 | Guinea | 2016 | Bahamas, The | 2018 | Sierra Leone | 2018 |
| Bhutan | 2012 | Kiribati | 2016 | Bahrain | 2018 | Solomon Islands | 2018 |
| Dominica | 2012 | Algeria | 2017 | Central African Republic | 2018 | Sudan | 2018 |
| Papua New Guinea | 2012 | Cameroon | 2017 | Dominican Republic | 2018 | Tanzania | 2018 |
| Turks and Caicos Islands | 2012 | Cote d'Ivoire | 2017 | Ethiopia | 2018 | Uganda | 2018 |
| | | East Timor | 2017 | Greenland | 2018 | Ukraine | 2018 |

## 5. Results

The datasets were inserted in the open software GEPHI for analyzing complex networks [20]. From the data, we specified the data matrix (Section 3), and we computed the indices (Sections 3.1–3.8).

### 5.1. Density, Degree, Clustering, and Modularity

The results for the density (Section 3.1), the degrees (Section 3.2), the clustering (Section 3.3), the modularity, and the identified communities (Section 3.4) including the unweighted graph are summarized in Table 2. The column data are presented for each year in Figures 2–4. We present in Figure 5 the variation of GDP (USD) during the years 2008–2019 for comparison. In Figure 6, the (out) flow index of ten important countries in the world trade is presented. The annual variations of modularity and the identified communities of Table 2 are visualized in Figure 7.

**Table 2.** Results of measurements of topological analysis of networks for 12 years. Nodes with no link in some years do not appear.

| Year | Nodes | Edges | Density | Avg Degree | Avg Weighted Degree | Avg Clustering Coefficient | Modularity (Weighted) | No. of Communities | | Modularity (Unweighted Graph) | No. of Communities |
|---|---|---|---|---|---|---|---|---|---|---|---|
| | | | | | | | | Case 1 | Case 2 | | |
| 2008 | 235 | 22,540 | 0.410 | 95.915 | 66,543,933 | 0.737 | 0.372 | 4 | 5 | 0.073 | 3 |
| 2009 | 236 | 22,497 | 0.406 | 95.326 | 51,379,623 | 0.733 | 0.363 | 5 | 5 | 0.073 | 4 |
| 2010 | 237 | 23,204 | 0.415 | 97.907 | 63,215,973 | 0.739 | 0.373 | 5 | 5 | 0.074 | 3 |
| 2011 | 238 | 22,946 | 0.407 | 96.412 | 75,004,671 | 0.741 | 0.379 | 5 | 5 | 0.071 | 4 |
| 2012 | 238 | 22,818 | 0.405 | 95.874 | 75,145,707 | 0.738 | 0.371 | 5 | 6 | 0.069 | 3 |
| 2013 | 239 | 23,176 | 0.407 | 96.971 | 77,928,476 | 0.746 | 0.383 | 5 | 5 | 0.071 | 3 |
| 2014 | 238 | 23,098 | 0.409 | 97.050 | 78,185,259 | 0.746 | 0.380 | 5 | 5 | 0.071 | 3 |
| 2015 | 238 | 23,466 | 0.416 | 98.597 | 68,164,456 | 0.751 | 0.354 | 5 | 6 | 0.071 | 3 |
| 2016 | 239 | 23,492 | 0.413 | 98.293 | 65,917,133 | 0.751 | 0.350 | 4 | 5 | 0.072 | 3 |
| 2017 | 238 | 23,409 | 0.415 | 98.357 | 73,012,586 | 0.753 | 0.373 | 4 | 5 | 0.072 | 3 |
| 2018 | 238 | 22,613 | 0.401 | 95.013 | 80,075,605 | 0.750 | 0.378 | 4 | 5 | 0.072 | 3 |
| 2019 | 238 | 20,045 | 0.355 | 84.223 | 76,664,339 | 0.743 | 0.362 | 4 | 5 | 0.072 | 3 |

We observed that the number of active nodes is almost stable, especially from 2011 onwards, as is shown in Figure 2 (black line), that is, approximately 238–239 nodes.

The approximate stability of the average degree (Figure 4, black line) is a manifestation of the globalization of the economy. The average degree slightly increased from 2008 to 2017, with an average value of 97 per year with fluctuations in the interval [−1.5, +1.5]. We did not take into account the years 2018–2019 because the extreme values were considered as outliers. The average weighted degree is presented in the red line, showing in absolute scale the sum of exports (weighted-out degree) of every country in thousands of USD. From the form of this specific curve, we clearly observe the following:

1. The recession of the world economy in the years of 2008 and 2009, which was instigated by the economic domino of the collapse of the American banks;

2.    The minor recession in the years 2014 to 2016, with the decrease in China's exports of 8% (2016);

3.    The upward trend during the development periods approaching USD 80 billion, with a consistent slope (2018).

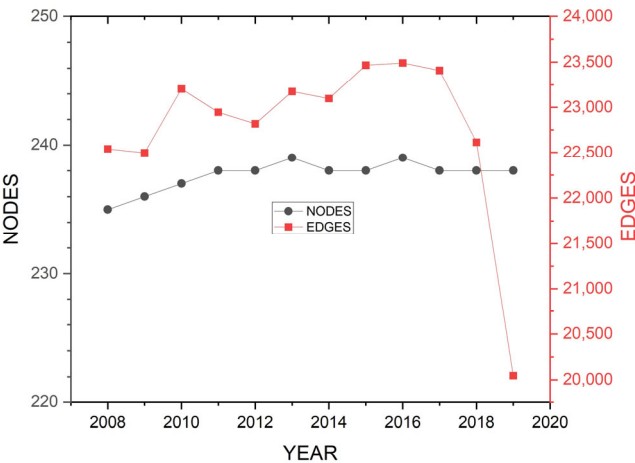

**Figure 2.** The number of nodes (black line) and the number of edges (red line).

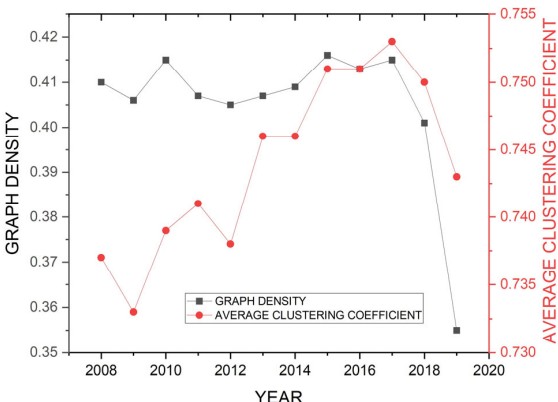

**Figure 3.** The variation of network density shown with the black line and the average clustering coefficient for the period 2008–2019 with the red line.

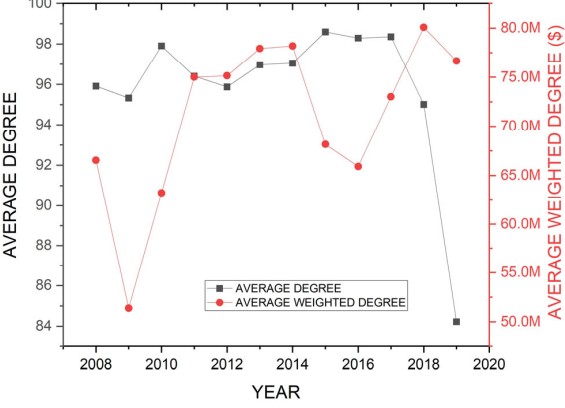

**Figure 4.** The average degree of nodes without weights (black line) and with weights (red line).

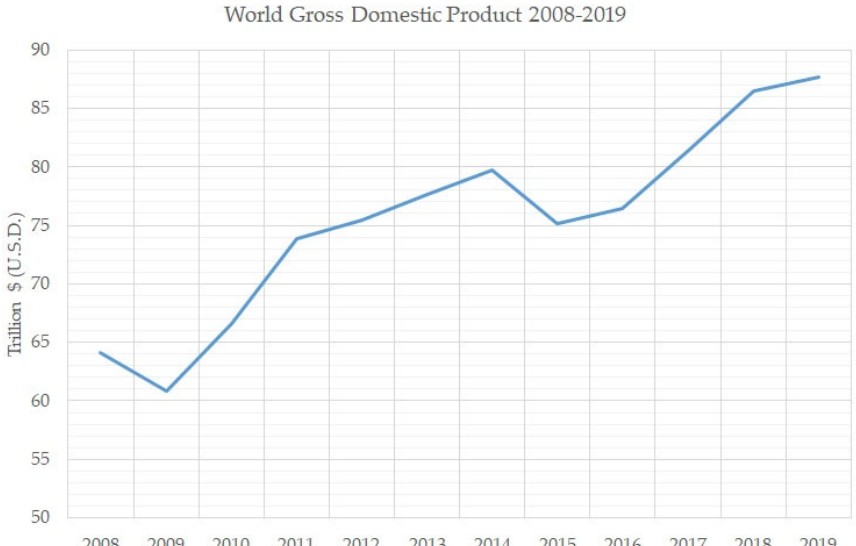

**Figure 5.** The variation of GDP in USD in the years 2008–2019. The local minimums are present in the same periods with the minimums in the average weighted degree indicator. Data source: World Development Indicators—The World Bank [25].

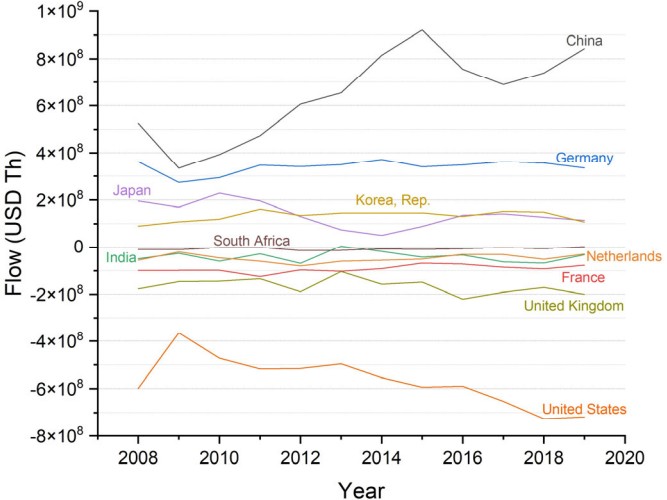

**Figure 6.** The (out) flow index for ten important countries.

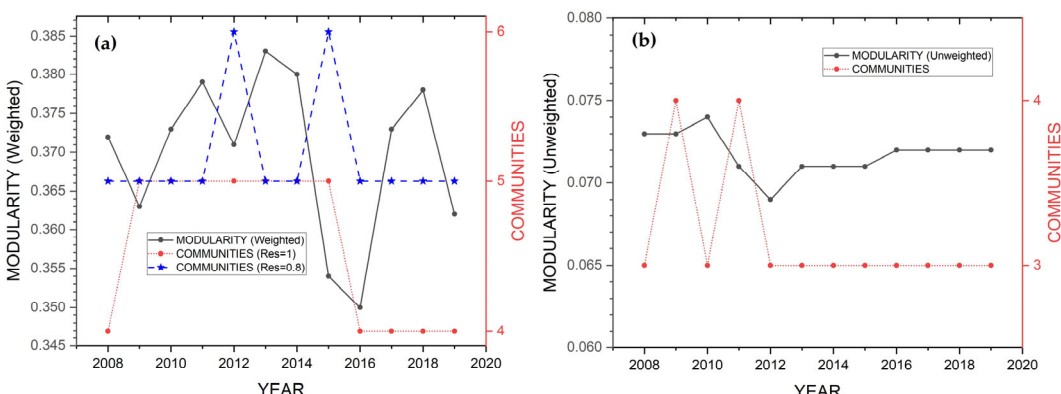

**Figure 7.** Fluctuation of modularity from 2008–2019 (black line, (**a**)). The red and blue lines show the differences in the number of communities present, modifying the algorithm (case 1–case 2, Table 2). The unweighted modularity is presented in (**b**) for comparison.

The above observations were also confirmed by the chart of world GDP (Figure 5), part of which depends on the countries' exports. The curve in Figure 5 is qualitatively similar to the red curve of the average weighted degree (Figure 4, red line).

The shape of the average weighted degree curve (Figure 4, red line) indicates that the lack of data in the years 2018 and 2019 does not have any significant influence. This is due to the small volume of external flows of the countries (Table 1).

We observed that the difference of exports from imports (Figure 6), also known as the balance of trade (B.o.T.), is close to zero for South Africa, India, and the Netherlands; is positive for Germany and most countries from Asia (China, Korea, and Japan); and is negative for the USA, the U.K., and France. Moreover, the USA and China evolved in the opposite way. In the year 2009, the USA and China presented opposite local extremes. In addition, the evolution of China (Figure 6) is very similar to the evolution of the world GDP (Figure 5).

The density (Figure 3, black line) shows stability around the value 0.41 with a 14% decrease in the last two years. The density changes (Figure 3, black line) are qualitatively similar to the changes of the average degree (Figure 4, black line).

The density fluctuations are less than 3%. This remarkable stability until 2018 indicates that the world economic network changes very slowly and that it is very robust to local changes.

The clustering coefficient was calculated separately for every node and represents the presence of links among the neighbors of each node of a network. The average clustering coefficient (Figure 3, red line) increased from 2008 to 2017 by 2.2% and afterwards decreased in the next year by 1.3%.

The simultaneous decrease in the average clustering coefficient (Figure 3, red line) and the average weighted degree (Figure 4, red line) in the year 2009 coincided with the decrease in global GDP. (Figure 5).

The modularity values in Table 2 for the networks from 2008 to 2019 indicate small changes (min–max: 0.35–0.383) around the average value of 0.37, as is visualized with the black line in Figure 7a. The modularity value indicates the presence of communities in the network. We selected two different criteria in the realization of the algorithm ([20]; resolution = 1, resolution = 0.8) for the termination of the algorithm (Figure 7a, red line and blue line) in order to identify possible additional communities, which are mostly four to five for the first criterion and five or six for the second criterion [17,26].

The unweighted modularity calculations present very stable values around 0.072 (Figure 7b, black line) and divide the networks in three or four groups (Figure 7b, red line). As this value of modularity is very close to zero, the community analysis of the unweighted graph is not significantly better than random [27]. Therefore, the analysis of the unweighted graph of the WTN is not appropriate.

*5.2. Members of the Groups*

The positioning of a country in a community is based on the number of commercial annual in and out flows. Since some countries show stability and uniformity in their transactions, they are consistently placed in the same community. However, there are other cases of countries that present volatile annual economic flows, and thus, they may change communities. We observed the formation of very few powerful blocks, while some other countries may change blocks every year. Table 3 shows these results for every year from 2008 to 2019. At the top of each column, we placed the dominant country of each community based on the volume of exports. In the last line, the average of the countries in each group was calculated so that the variations in their numbers are easier to perceive.

Possible reasons for the presence of a country in a community include the following:

1. Its geographic proximity to the rest of the countries obviously leads to the reduction of the transportation cost (e.g., Baltic Sea countries, countries of North America, etc.);
2. The political proximity among countries leads to mutual help and the formation of trade relationships (UK–Australia; China–North Korea);

3. The social and cultural proximity among peoples create stable bonds over the years (Turkey–Azerbaijan; Greece–Cyprus);
4. The international economic circumstances lead to a change in the inflows and outflows of a country (currency devaluation and purchase of oil and raw materials);
5. The hegemonic presence of a country in a community brings about a feeling of security to the rest of the countries that belong to this community (USA, Germany, and China) [28].

**Table 3.** The communities and the number of their member countries based on modularity analysis (criterion 1, resolution = 1, GEPHI [20]).

| Year | Germany | China | USA | S. Arabia/UAE | S. Africa/India | Africa 2 | SUM |
|------|---------|-------|-----|---------------|-----------------|----------|-----|
| 2008 | 76 | 86 | 54 | 19 | | | 235 |
| 2009 | 78 | 84 | 57 | 13 | | 4 | 236 |
| 2010 | 74 | 65 | 63 | 14 | 21 | | 237 |
| 2011 | 86 | 62 | 59 | 17 | 14 | | 238 |
| 2012 | 73 | 55 | 38 | 17 | 55 | | 238 |
| 2013 | 82 | 67 | 56 | 15 | 19 | | 239 |
| 2014 | 77 | 68 | 65 | 14 | 14 | | 238 |
| 2015 | 75 | 64 | 66 | 14 | 19 | | 238 |
| 2016 | 76 | 82 | 40 | 41 | | | 239 |
| 2017 | 73 | 52 | 50 | 63 | | | 238 |
| 2018 | 77 | 87 | 50 | 24 | | | 238 |
| 2019 | 77 | 98 | 46 | 17 | | | 238 |
| *AVG* | *77* | *73* | *54* | *22* | *24* | *4* | *238* |

Figure 8a shows the communities and the number of countries that belong to them, are presented with a different color, from 2008 to 2019 for the first criterion of the termination of the modularity algorithm. Each community takes the name of the country with the greatest volume of exports:

1. Germany (black curve);
2. China (red curve);
3. USA (blue curve);
4. Arab countries (Saudi Arabia/United Arab Emirates—green curve).

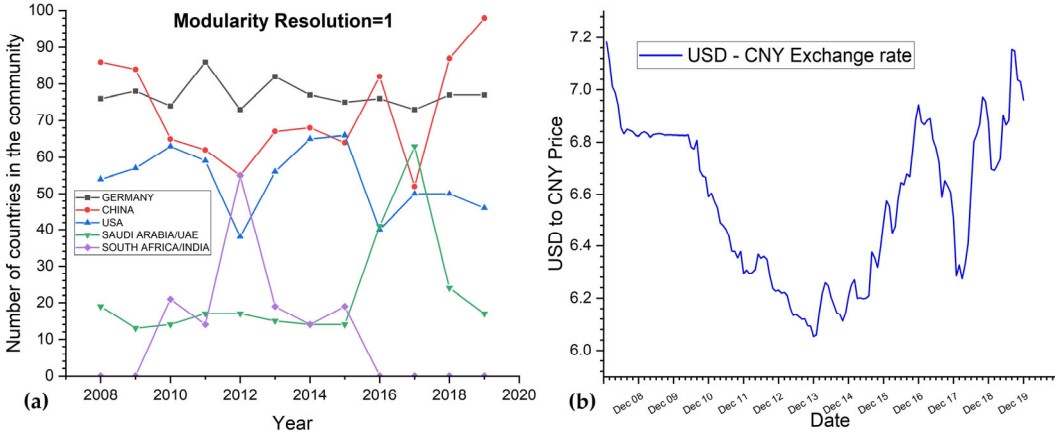

**Figure 8.** (**a**) The modularity analysis in the first scenario produces three main communities and two minor ones. (**b**) The exchange rate indicator USD–CNY. Data source: World Integrated Trade Solution—The World Bank [24].

Furthermore, from 2010 to 2015, the South African group appears with a small number of members (purple curve), whereas for 2012, India is added, too. Finally, the four African countries, which appear only in 2009 in Table 3, cannot be considered as a separate community.

As is shown in the last column (SUM) of Table 3, the number of countries that participate in the creation of communities is stable. Therefore, the increase in members in one group implies the decrease in members in some other group. The conclusions that arise from the comparison of curves are the following:

1.  The stability of Germany's community, with a group of 77 countries on average, is due to the powerful economic connection of European Union's countries and other countries of the broader geographic area (Sweden, Norway, and Switzerland). We can also observe that the curve's ranges from 2010 to 2014 coincide with the period of the crisis of 2008 exit and the beginning of debt crisis in Greece;
2.  China influences global economy with the creation of the largest group of countries, which ranges widely from 50 to 100 countries. These changes are due to the volatile conditions that prevail in the world economy caused by strong competition and relate to periods of significant political and financial events, such as the imposition of duties, exchange rates, and oil prices;
3.  The third largest group in member countries (36–66 countries) belong to the sphere of economic influence of the USA and also show considerable ranges. We observe (Figure 8a) that the number of member countries in the group of the USA and the number of member countries in the group of China evolve in the opposite way. If we compare the curves of China–USA, we notice the reverse course for each one of them (with the exception of 2012), which clearly demonstrates the competition between the two groups over the recruitment of more member countries. The transition of countries from one group to the other is mainly due to financial factors such as the exchange rate USD–CNY, which is presented for the years 2008–2019 in Figure 8b. Comparing Figure 8a,b, it becomes obvious that the more powerful the USD is over the CNY (2008–2010, 2016, and 2018–2019), the more countries join the group of China. Conversely, from 2012 to 2015, the higher exchange rate of the Chinese currency gave the rest of the groups the ability to raise the number of their member countries;
4.  The community of the oil-producing countries of the Gulf has a stable number of member countries (13–19 countries) from the beginning of this investigation until January 2016, when the oil reached its lowest price for the period 2008–2019, which ultimately led to growing demand. Thus, at this point, there was a soaring increase in the member countries of the group, reaching 63 in 2017, and in the next two years, this number approximated the initial number;
5.  Finally, the community of African countries appeared from 2010 to 2015 and, with the exception of 2012, had from 14 to 21 member countries. In this small group, the country with the biggest export turnover is South Africa. However, for some years and specifically in 2012, other countries were included, such as India, Brazil, Argentina, Nigeria, Pakistan, and many more, thus temporarily creating a large group of 55 member countries. It is remarkable that Brazil, India, and (from 2010) South Africa were three out of the five members of the "BRICS" group, and moreover, the rest of the countries have shown their interest to join this particular group [29].

Figures 9a and 10a show the communities within the complex trade networks for two representative years (2008, 2019) based on the calculation of modularity with the first criterion. In these networks, the communities are presented in a different color, while the size of each node is proportional to the exports of the country it represents. In addition, the *OpenOrd* algorithm was used to place the nodes in the specific positions in each community [30]. We present in Figures 9b and 10b the communities for the unweighted graph for comparison.

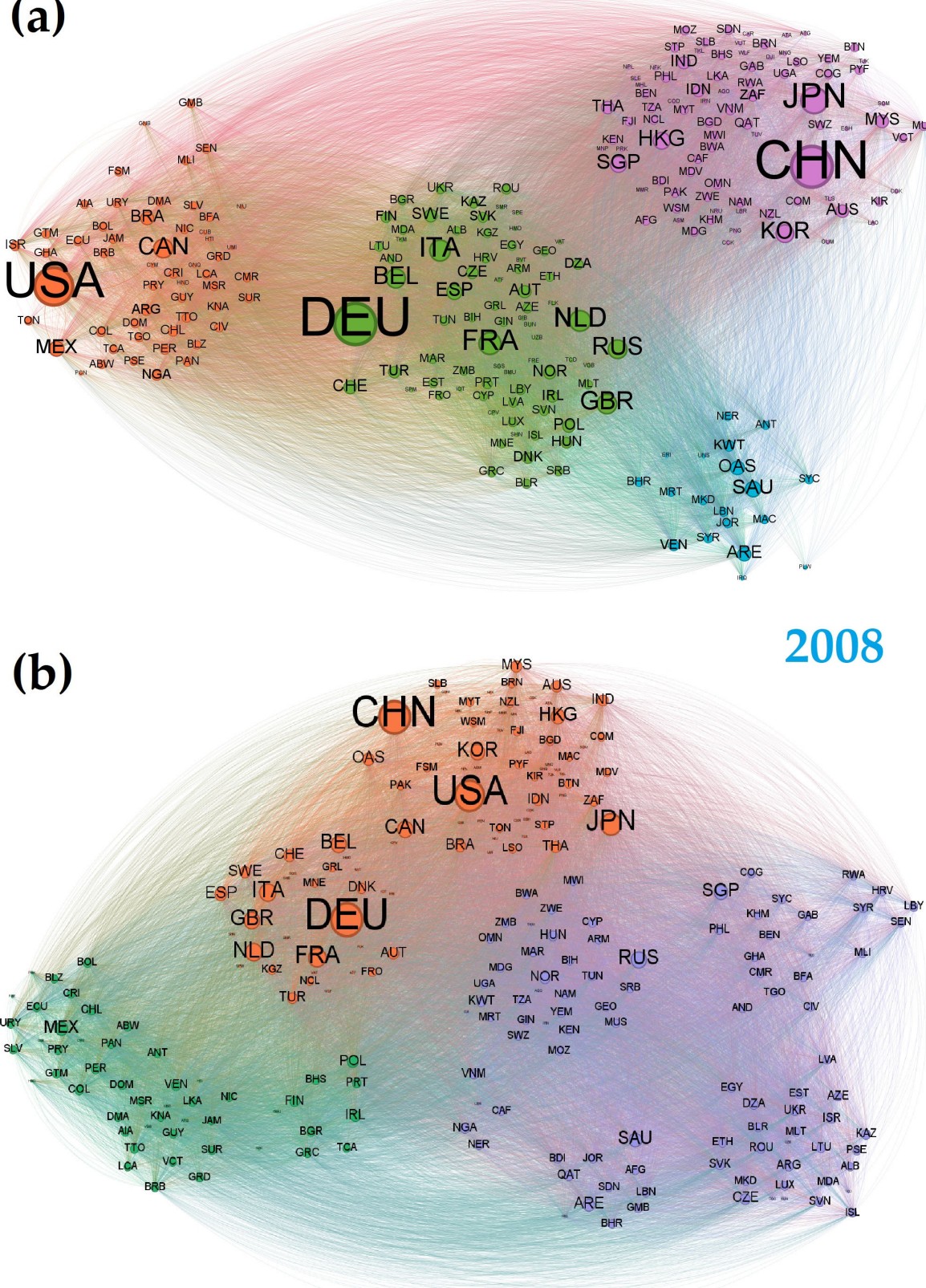

**Figure 9.** The community structure of the WTN for the year 2008 involves four communities with the key players of the USA, China, Germany, and S. Arabia/UAE (**a**). The community structure for the corresponding unweighted graph involves three communities with mixed key players (**b**).

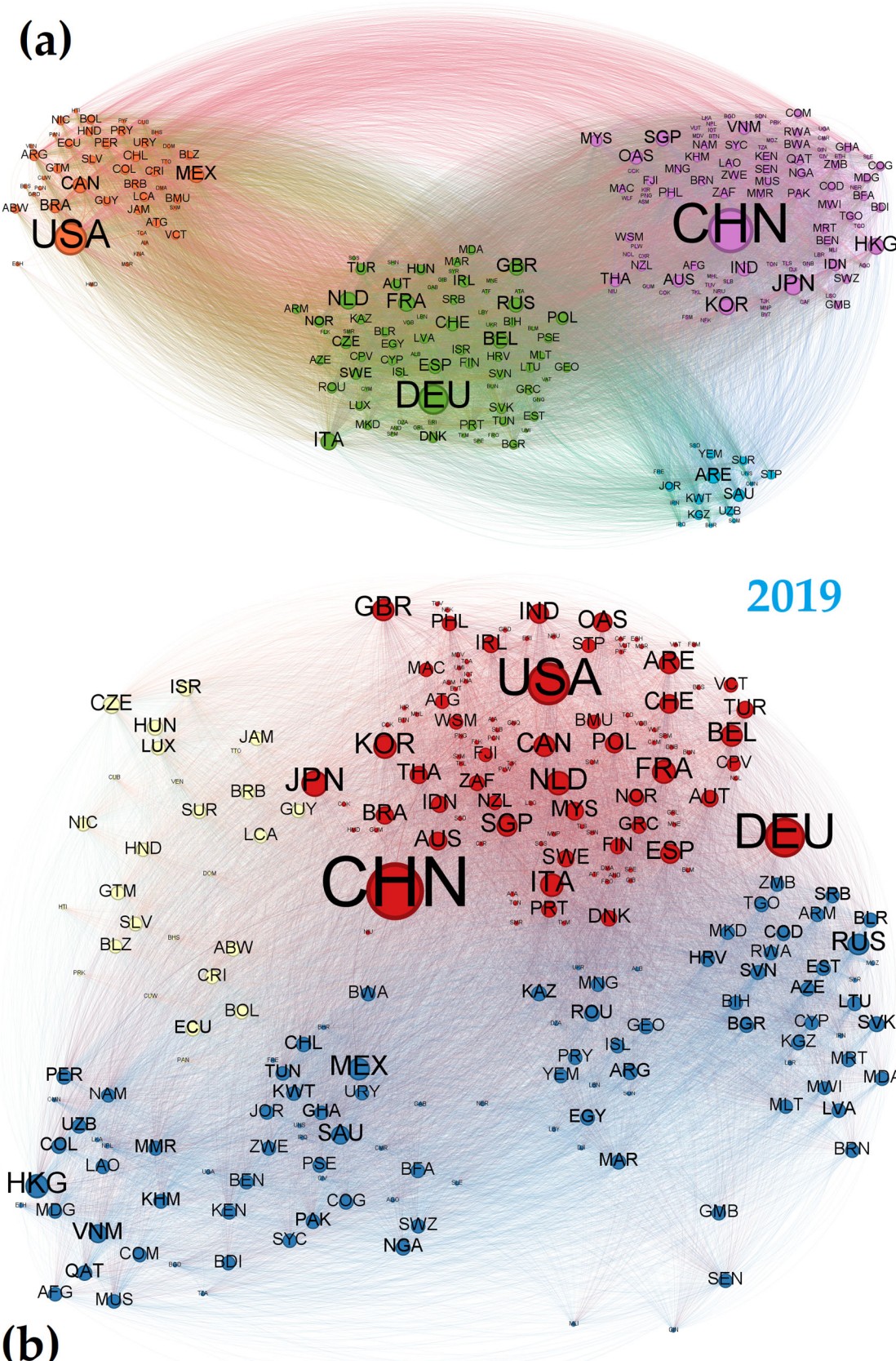

**Figure 10.** (**a**) The community structure of the WTN for the year 2019 involves four communities with the key players of the USA, China, Germany, and S. Arabia/UAE (**a**). The community structure for the corresponding unweighted graph involves three communities with mixed key players (**b**).

We observed that there was a significant difference between the community analysis of the WTN and the associated unweighted graph in both years 2008 was 2019. More specifically, the community structure of the WTN involves four communities with the key players of the USA, China, Germany, and Arab countries, while the corresponding unweighted graph involves three communities with mixed key players (Figures 9 and 10). Similar differences between WTN and the corresponding unweighted graph appeared in all years. In the years 2009–2015 appear five communities for the WTN (Table 2) with the key players of the USA, Germany, China, S. Arabia/UAE, and S. Africa/India (Table 3).

In order to examine the impact of economic flows on geopolitical relationships, we refined the community analysis, modifying the termination criterion (resolution = 0.8). In Table 4, we present the results in the same way as in Table 3.

**Table 4.** The numerical distribution of countries in different communities in the second scenario about modularity (criterion 2, resolution = 0.8, GEPHI [20]).

| Year | Germany | China | USA | S. Arabia/UAE | S. Africa/India | Russia | UK | *SUM* |
|------|---------|-------|-----|---------------|-----------------|--------|-----|-------|
| 2008 | 64 | 51 | 40 | 14 | 66 | | | 235 |
| 2009 | 83 | 42 | 41 | 14 | 56 | | | 236 |
| 2010 | 70 | 83 | 49 | 13 | 22 | | | 237 |
| 2011 | 67 | 51 | 44 | 10 | 66 | | | 238 |
| 2012 | 50 | 46 | 34 | 19 | 51 | 38 | | 238 |
| 2013 | 70 | 106 | 40 | 15 | | | 8 | 239 |
| 2014 | 52 | 108 | 42 | 2 | | 34 | | 238 |
| 2015 | 55 | 75 | 35 | 11 | 10 | 52 | | 238 |
| 2016 | 62 | 55 | 65 | 32 | 25 | | | 239 |
| 2017 | 75 | 47 | 38 | 17 | 61 | | | 238 |
| 2018 | 72 | 61 | 57 | 30 | 18 | | | 238 |
| 2019 | 72 | 74 | 52 | 27 | 13 | | | 238 |
| AVG | 66 | 67 | 45 | 17 | 39 | 41 | 8 | 238 |

Comparing Tables 3 and 4, we observe a decrease in the number of countries in the four main groups (from −8% to −23%). On the other hand, there is an increase in the member countries in South Africa's/India's community (+63%). Communities that appear for the first time are those of Russia, with many of the Former Soviet Republics for the period of three years, and once the United Kingdom as a separate group. Figure 11a shows the three communities with the largest number of member countries, and Figure 11b shows the rest of them.

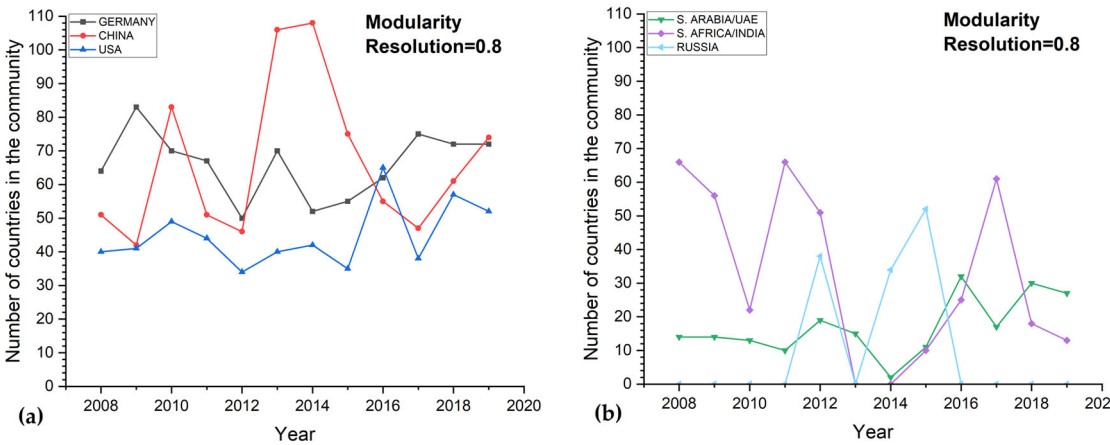

**Figure 11.** (**a**) The changes in the number of member countries in the three main communities (Germany, the USA, and China) are greater due to the appearance of minor communities (**b**) with more member countries (Saudi Arabia/HAE, South Africa/India, and Russia).

Figure 12 shows the curves with those countries whose changes are more relevant.

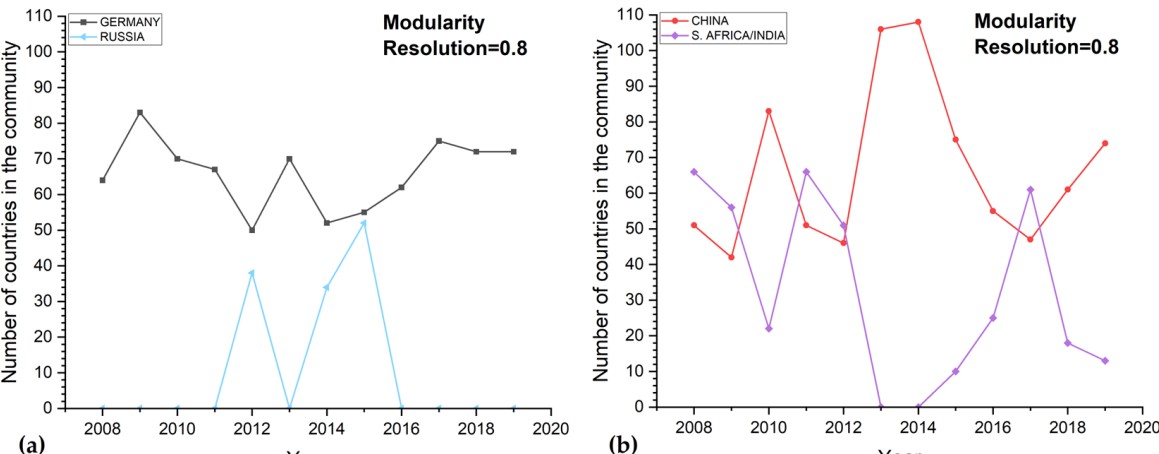

**Figure 12.** (**a**) Fluctuations in the stable group of Germany appear whenever Russia's group occasionally appears. (**b**) The negative interrelation and competition between China–India/South Africa.

The group of Germany (Figure 12a, black line) is more affected than the rest of the main communities by the appearance of the group of Russia (Figure 12a, blue line). The presence of the India–South Africa group (Figure 12b, purple line) brings about more variation in the China community (Figure 12b, red line).

The community analysis of the world trade with the second criterion (resolution = 0.8) shows the following:

1.  Germany's community is influenced more by the sporadic autonomy of Russia, which is followed by several countries (34 to 52) due to their close relationships (Ukraine, Baltic Countries, Turkey, Egypt, and Jordan);
2.  China and South Africa/India compete with each other more than any other community as far as the number of their community's member countries is concerned, which showcases the uncertainty and dynamics in this part of the world economy;
3.  The USA remains third, and there is no tendency of separation in the group in which they have a hegemonic role;
4.  In 2013, a small independent community (eight members) appeared, including the United Kingdom, Ireland, Switzerland, and some smaller countries (Zambia, Lebanon, etc.). In this group, two of the most powerful financial/banking countries are connected, with one of them (the U.K.) being a member of the European Union at that time.

### 5.3. Distance and Topology

The computation of the diameter, the out eccentricity, and average path length of the network is presented in Table 5. The annual variations of the average path length are visualized in Figure 13.

Each country is connected to any other country by two or three steps, and the average path length changes by 5.4%. The robustness of these indicators are a manifestation of the high global interdependence. This small but noticeable decrease in the average path length (Figure 13) is understood from the increase in the average clustering coefficient in the same period (Figure 3, red line), which are both manifestations of the growing trend towards globalization.

**Table 5.** Geometrical analysis. The third column presents the minimum and the maximum values of out eccentricity observed during each year.

| Year | Diameter | Out Eccentricity of Nodes (Min–Max) | Average Path Length |
|------|----------|-------------------------------------|---------------------|
| 2008 | 3 | 2–3 | 1.437 |
| 2009 | 3 | 2–3 | 1.442 |
| 2010 | 3 | 2–3 | 1.430 |
| 2011 | 3 | 2–3 | 1.428 |
| 2012 | 3 | 2–3 | 1.434 |
| 2013 | 3 | 2–3 | 1.417 |
| 2014 | 3 | 2–3 | 1.420 |
| 2015 | 3 | 2–3 | 1.407 |
| 2016 | 3 | 2–3 | 1.403 |
| 2017 | 3 | 2–3 | 1.390 |
| 2018 | 3 | 2–3 | 1.359 |
| 2019 | 3 | 2–3 | 1.360 |

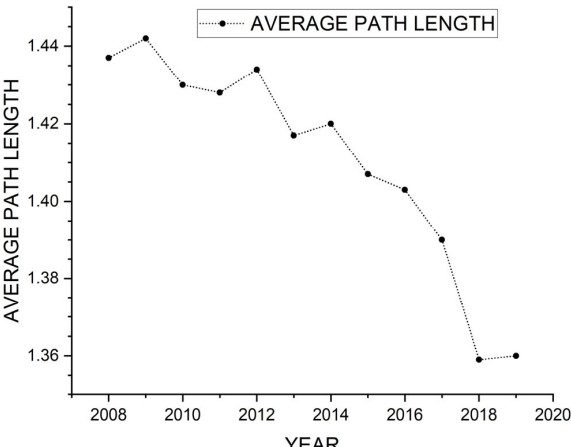

**Figure 13.** The decrease in the average path length.

## 5.4. Closeness Centrality

The annual closeness centralities and the average values are presented in Table 6. The annual relative increase in closeness centrality is presented in Table 7, and the fluctuations in closeness centrality for ten selected countries are presented in Figure 14.

**Table 6.** Annual closeness centrality of the 18 countries with highest values. The five countries with highest centrality are indicated in light red.

| Country/Year | 2008 | 2009 | 2010 | 2011 | 2012 | 2013 | 2014 | 2015 | 2016 | 2017 | 2018 | 2019 | AVG |
|--------------|------|------|------|------|------|------|------|------|------|------|------|------|-----|
| Germany | 0.98734 | 0.98326 | 0.97119 | 0.94800 | 0.95565 | 0.97143 | 0.97131 | 0.97131 | 0.97942 | 0.97131 | 0.97131 | 0.97934 | 0.97174 |
| U.K. | 0.97908 | 0.97510 | 0.97119 | 0.96342 | 0.94800 | 0.96748 | 0.97131 | 0.97531 | 0.96748 | 0.97531 | 0.97531 | 0.96342 | 0.96937 |
| France | 0.97500 | 0.96312 | 0.95935 | 0.94800 | 0.93676 | 0.96748 | 0.96342 | 0.96342 | 0.96356 | 0.97131 | 0.95951 | 0.95565 | 0.96055 |
| Netherlands | 0.97500 | 0.95918 | 0.96327 | 0.96342 | 0.96735 | 0.98755 | 0.97531 | 0.98340 | 0.98755 | 0.97934 | 0.98750 | 0.98750 | 0.97636 |
| Belgium | 0.97500 | 0.97107 | 0.95161 | 0.95565 | 0.96342 | 0.97541 | 0.97131 | 0.95951 | 0.95582 | 0.97131 | 0.97131 | 0.96735 | 0.96573 |
| Switzerland | 0.96694 | 0.94758 | 0.94779 | 0.94800 | 0.95181 | 0.95968 | 0.95565 | 0.95181 | 0.94071 | 0.95951 | 0.96735 | 0.94048 | 0.95311 |
| United States | 0.95902 | 0.95529 | 0.94779 | 0.95181 | 0.95565 | 0.94821 | 0.95181 | 0.94800 | 0.94821 | 0.94800 | 0.94800 | 0.94422 | 0.95050 |
| Spain | 0.95902 | 0.92885 | 0.94779 | 0.94048 | 0.93676 | 0.95582 | 0.94800 | 0.95565 | 0.96748 | 0.96342 | 0.94422 | 0.95951 | 0.95058 |
| Denmark | 0.95510 | 0.94758 | 0.95161 | 0.92578 | 0.92218 | 0.95582 | 0.94422 | 0.94800 | 0.94444 | 0.94048 | 0.94422 | 0.94048 | 0.94333 |
| Malaysia | 0.95510 | 0.94758 | 0.95161 | 0.94422 | 0.95565 | 0.94821 | 0.93307 | 0.95181 | 0.91892 | 0.92941 | 0.94048 | 0.91861 | 0.94122 |
| Italy | 0.95122 | 0.94758 | 0.95161 | 0.95565 | 0.94422 | 0.97143 | 0.95565 | 0.95951 | 0.95968 | 0.97131 | 0.95565 | 0.97934 | 0.95857 |
| India | 0.95122 | 0.94758 | 0.94779 | 0.93307 | 0.93676 | 0.93333 | 0.93676 | 0.93307 | 0.92607 | 0.92941 | 0.94048 | 0.95565 | 0.93927 |
| Other Asia, nes | 0.95122 | 0.95142 | 0.94400 | 0.94422 | 0.94048 | 0.92248 | 0.92218 | 0.90805 | 0.91539 | 0.91506 | 0.93676 | 0.91861 | 0.93082 |
| Sweden | 0.94737 | 0.55529 | 0.94400 | 0.94800 | 0.91861 | 0.94444 | 0.94800 | 0.94048 | 0.97541 | 0.95951 | 0.96342 | 0.94800 | 0.94938 |
| Korea, Rep. | 0.93976 | 0.55529 | 0.94400 | 0.95181 | 0.95181 | 0.94821 | 0.94048 | 0.93307 | 0.93333 | 0.94422 | 0.94422 | 0.93676 | 0.94358 |
| Canada | 0.93600 | 0.93626 | 0.92549 | 0.91506 | 0.92218 | 0.91188 | 0.94048 | 0.92218 | 0.92969 | 0.94048 | 0.92941 | 0.92941 | 0.92821 |
| Australia | 0.93600 | 0.92157 | 0.91829 | 0.90458 | 0.90805 | 0.89811 | 0.90458 | 0.90458 | 0.89811 | 0.90458 | 0.90805 | 0.90114 | 0.90897 |
| Thailand | 0.93227 | 0.55529 | 0.93281 | 0.93307 | 0.94422 | 0.94071 | 0.96342 | 0.96342 | 0.95200 | 0.95565 | 0.96735 | 0.95565 | 0.94965 |
| AVG | 0.95731 | 0.95272 | 0.94840 | 0.94301 | 0.94220 | 0.95043 | 0.94983 | 0.94848 | 0.94796 | 0.95165 | 0.95303 | 0.94895 | |

**Table 7.** Annual relative increase in closeness centrality for 18 selected countries. Each column gives the relative increase with respect to the previous year. Negative values indicate decrease.

| Country/Year | 2008 (Cc) | 2009 (%) | 2010 (%) | 2011 (%) | 2012 (%) | 2013 (%) | 2014 (%) | 2015 (%) | 2016 (%) | 2017 (%) | 2018 (%) | 2019 (%) |
|---|---|---|---|---|---|---|---|---|---|---|---|---|
| Germany | 0.98734 | −0.4 | −1.2 | −2.4 | 0.8 | 1.7 | 0.0 | 0.0 | 0.8 | −0.8 | 0.0 | 0.8 |
| U.K. | 0.97908 | −0.4 | −0.4 | −0.8 | −1.6 | 2.1 | 0.4 | 0.4 | −0.8 | 0.8 | 0.0 | −1.2 |
| France | 0.97500 | −1.2 | −0.4 | −1.2 | −1.2 | 3.3 | −0.4 | 0.0 | 0.0 | 0.8 | −1.2 | −0.4 |
| Netherlands | 0.97500 | −1.6 | 0.4 | 0.0 | 0.4 | 2.1 | −1.2 | 0.8 | 0.4 | −0.8 | 0.8 | 0.0 |
| Belgium | 0.97500 | −0.4 | −2.0 | 0.4 | 0.8 | 1.2 | −0.4 | −1.2 | −0.4 | 1.6 | 0.0 | −0.4 |
| Switzerland | 0.96694 | −2.0 | 0.0 | 0.0 | 0.4 | 0.8 | −0.4 | −0.4 | −1.2 | 2.0 | 0.8 | −2.8 |
| United States | 0.95902 | −0.4 | −0.8 | 0.4 | 0.4 | −0.8 | 0.4 | −0.4 | 0.0 | 0.0 | 0.0 | −0.4 |
| Spain | 0.95902 | −3.1 | 2.0 | −0.8 | −0.4 | 2.0 | −0.8 | 0.8 | 1.2 | −0.4 | −2.0 | 1.6 |
| Denmark | 0.95510 | −0.8 | 0.4 | −2.7 | −0.4 | 3.6 | −1.2 | 0.4 | −0.4 | −0.4 | 0.4 | −0.4 |
| Malaysia | 0.95510 | −0.8 | 0.4 | −0.8 | 1.2 | −0.8 | −1.6 | 2.0 | −3.5 | 1.1 | 1.2 | −2.3 |
| Italy | 0.95122 | −0.4 | 0.4 | 0.4 | −1.2 | 2.9 | −1.6 | 0.4 | 0.0 | 1.2 | −1.6 | 2.5 |
| India | 0.95122 | −0.4 | 0.0 | −1.6 | 0.4 | −0.4 | 0.4 | −0.4 | −0.8 | 0.4 | 1.2 | 1.6 |
| Other Asia, nes | 0.95122 | 0.0 | −0.8 | 0.0 | −0.4 | −1.9 | 0.0 | −1.5 | 0.8 | 0.0 | 2.4 | −1.9 |
| Sweden | 0.94737 | 0.8 | −1.2 | 0.4 | −3.1 | 2.8 | 0.4 | −0.8 | 3.7 | −1.6 | 0.4 | −1.6 |
| Korea, Rep. | 0.93976 | 1.7 | −1.2 | 0.8 | 0.0 | −0.4 | −0.8 | −0.8 | 0.0 | 1.2 | 0.0 | −0.8 |
| Canada | 0.93600 | 0.0 | −1.1 | −1.1 | 0.8 | −1.1 | 3.1 | −1.9 | 0.8 | 1.2 | −1.2 | 0.0 |
| Australia | 0.93600 | −1.5 | −0.4 | −1.5 | 0.4 | −1.1 | 0.7 | 0.0 | −0.7 | 0.7 | 0.4 | −0.8 |
| Thailand | 0.93227 | 2.5 | −2.4 | 0.0 | 1.2 | −0.4 | 2.4 | 0.0 | −1.2 | 0.4 | 1.2 | −1.2 |

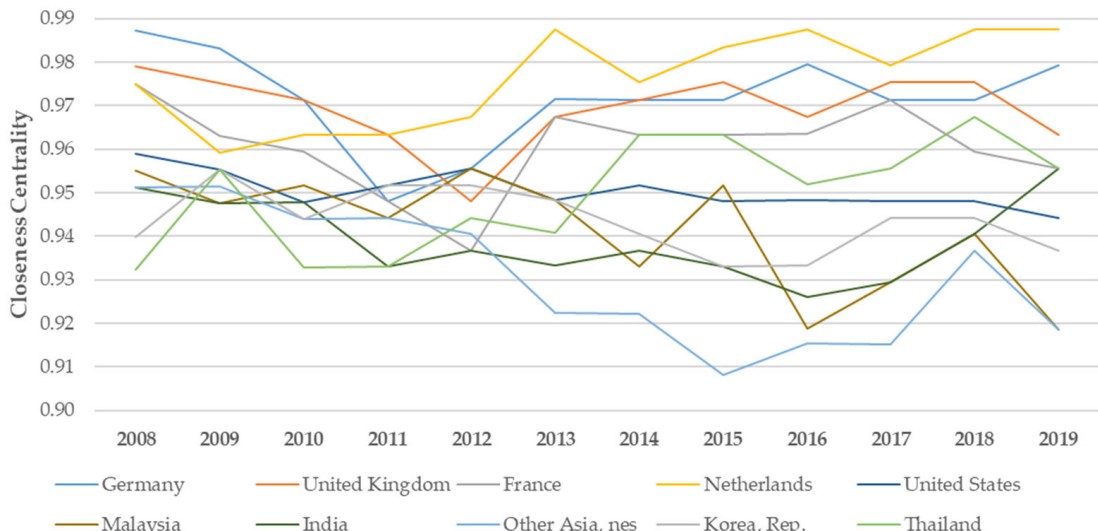

**Figure 14.** The fluctuations in closeness centrality for 10 important countries in the world trade with highest values.

Most dominant countries remained dominant from 2008 to 2019. The robustness of the average closeness over the years indicates that the structure of imports–exports among those countries was stable (Table 6). We observed (Table 7) that for the year 2009, there was a decrease in the relative values of $C_c$ for the Western economies and an increase in the values of the index for the Eastern world economies (Korea Rep. and Thailand), while the opposite was observed for the year 2013. This observation is related with the 2008–2009 crisis in the Western economy and the recovery from this crisis in 2013.

*5.5. Betweenness Centrality*

The annual betweenness centralities as well as the averages are presented in Table 8. The annual relative increase in betweenness centrality is presented in Table 9, and the fluctuations in betweenness centrality for ten selected countries are presented in Figure 15.

**Table 8.** Annual betweenness centrality of the 18 countries with highest values. The five countries with highest centrality are indicated in light red.

| Country/Year | 2008 | 2009 | 2010 | 2011 | 2012 | 2013 | 2014 | 2015 | 2016 | 2017 | 2018 | 2019 | AVG |
|---|---|---|---|---|---|---|---|---|---|---|---|---|---|
| United States | 0.01580 | 0.01565 | 0.01233 | 0.01639 | 0.01580 | 0.01227 | 0.00876 | 0.00861 | 0.00795 | 0.00856 | 0.00793 | 0.00583 | 0.01132 |
| U.K. | 0.01028 | 0.00934 | 0.00918 | 0.00827 | 0.00773 | 0.00840 | 0.00924 | 0.00841 | 0.00630 | 0.00689 | 0.00684 | 0.00541 | 0.00802 |
| France | 0.00957 | 0.00882 | 0.00773 | 0.00658 | 0.00709 | 0.00814 | 0.00887 | 0.00675 | 0.00818 | 0.00636 | 0.00578 | 0.00534 | 0.00743 |
| Germany | 0.00918 | 0.00995 | 0.00819 | 0.00790 | 0.00908 | 0.00868 | 0.00858 | 0.00794 | 0.00855 | 0.00683 | 0.00662 | 0.00521 | 0.00806 |
| Netherlands | 0.00824 | 0.00828 | 0.00804 | 0.00759 | 0.00799 | 0.00941 | 0.00770 | 0.00898 | 0.00817 | 0.00873 | 0.00890 | 0.00707 | 0.00826 |
| Belgium | 0.00756 | 0.00839 | 0.00612 | 0.00629 | 0.00665 | 0.00724 | 0.00740 | 0.00628 | 0.00599 | 0.00572 | 0.00596 | 0.00503 | 0.00655 |
| Canada | 0.00719 | 0.00831 | 0.00675 | 0.00598 | 0.00670 | 0.00476 | 0.00608 | 0.00659 | 0.00610 | 0.00588 | 0.00512 | 0.00512 | 0.00622 |
| Australia | 0.00716 | 0.00661 | 0.00620 | 0.00611 | 0.00579 | 0.00461 | 0.00602 | 0.00514 | 0.00453 | 0.00494 | 0.00446 | 0.00446 | 0.00550 |
| China | 0.00691 | 0.00706 | 0.00602 | 0.00650 | 0.00834 | 0.00586 | 0.00663 | 0.00623 | 0.00548 | 0.00534 | 0.00510 | 0.00483 | 0.00619 |
| Switzerland | 0.00690 | 0.00629 | 0.00673 | 0.00656 | 0.00763 | 0.00637 | 0.00575 | 0.00611 | 0.00639 | 0.00603 | 0.00555 | 0.00555 | 0.00632 |
| Japan | 0.00679 | 0.00786 | 0.00723 | 0.00554 | 0.00540 | 0.00488 | 0.00515 | 0.00522 | 0.00554 | 0.00467 | 0.00448 | 0.00448 | 0.00560 |
| Italy | 0.00669 | 0.00730 | 0.00674 | 0.00737 | 0.00695 | 0.00795 | 0.00667 | 0.00667 | 0.00650 | 0.00648 | 0.00604 | 0.00516 | 0.00671 |
| Other Asia, nes | 0.00654 | 0.00695 | 0.00686 | 0.00656 | 0.00631 | 0.00520 | 0.00511 | 0.00507 | 0.00496 | 0.00417 | 0.00505 | 0.00505 | 0.00565 |
| Spain | 0.00637 | 0.00536 | 0.00707 | 0.00582 | 0.00549 | 0.00606 | 0.00612 | 0.00638 | 0.00580 | 0.00556 | 0.00531 | 0.00516 | 0.00588 |
| Denmark | 0.00619 | 0.00572 | 0.00583 | 0.00499 | 0.00488 | 0.00593 | 0.00509 | 0.00509 | 0.00472 | 0.00482 | 0.00473 | 0.00320 | 0.00510 |
| India | 0.00589 | 0.00673 | 0.00686 | 0.00647 | 0.00726 | 0.00665 | 0.00620 | 0.00531 | 0.00464 | 0.00433 | 0.00498 | 0.00498 | 0.00586 |
| Korea, Rep. | 0.00547 | 0.00596 | 0.00644 | 0.00630 | 0.00682 | 0.00567 | 0.00632 | 0.00520 | 0.00609 | 0.00492 | 0.00537 | 0.00537 | 0.00583 |
| South Africa | 0.00506 | 0.00474 | 0.00593 | 0.00557 | 0.00458 | 0.00460 | 0.00514 | 0.00410 | 0.00381 | 0.00468 | 0.00413 | 0.00409 | 0.00470 |
| AVG | 0.00765 | 0.00774 | 0.00724 | 0.00704 | 0.00725 | 0.00682 | 0.00671 | 0.00634 | 0.00609 | 0.00583 | 0.00569 | 0.00507 | |

**Table 9.** Annual relative increase in betweenness centrality for 18 selected countries. Each column gives the relative increase with respect to the previous year. Negative values indicate decrease.

| Country/Year | 2008 (Cb) | 2009 (%) | 2010 (%) | 2011 (%) | 2012 (%) | 2013 (%) | 2014 (%) | 2015 (%) | 2016 (%) | 2017 (%) | 2018 (%) | 2019 (%) |
|---|---|---|---|---|---|---|---|---|---|---|---|---|
| United States | 0.01580 | −0.9 | −21.2 | 32.9 | −3.6 | −22.3 | −28.6 | −1.7 | −7.6 | 7.7 | −7.4 | −26.4 |
| U.K. | 0.01028 | −9.2 | −1.7 | −9.9 | −6.6 | 8.7 | 10.0 | −9.0 | −25.1 | 9.3 | −0.7 | −21.0 |
| France | 0.00957 | −7.9 | −12.3 | −14.9 | 7.7 | 14.9 | 8.9 | −23.9 | 21.2 | −22.2 | −9.2 | −7.5 |
| Germany | 0.00918 | 8.4 | −17.7 | −3.5 | 15.0 | −4.4 | −1.2 | −7.5 | 7.6 | −20.1 | −3.0 | −21.3 |
| Netherlands | 0.00824 | 0.6 | −2.9 | −5.6 | 5.3 | 17.7 | −18.1 | 16.6 | −9.1 | 6.8 | 2.0 | −20.6 |
| Belgium | 0.00756 | 11.1 | −27.1 | 2.8 | 5.6 | 9.0 | 2.2 | −15.1 | −4.6 | −4.6 | 4.3 | −15.5 |
| Canada | 0.00719 | 15.5 | −18.7 | −11.4 | 12.1 | −28.9 | 27.7 | 8.4 | −7.5 | −3.6 | −13.0 | 0.0 |
| Australia | 0.00716 | −7.6 | −6.2 | −1.3 | −5.3 | −20.4 | 30.5 | −14.5 | −11.9 | 9.0 | −9.8 | 0.0 |
| China | 0.00691 | 2.1 | −14.7 | 8.0 | 28.3 | −29.8 | 13.1 | −6.0 | −12.0 | −2.5 | −4.5 | −5.3 |
| Switzerland | 0.00690 | −8.9 | 6.9 | −2.4 | 16.2 | −16.5 | −9.7 | 6.1 | 4.7 | −5.6 | −7.9 | 0.0 |
| Japan | 0.00679 | 15.7 | −8.0 | −23.3 | −2.6 | −9.6 | 5.5 | 1.4 | 6.0 | −15.7 | −4.1 | 0.0 |
| Italy | 0.00669 | 9.1 | −7.7 | 9.3 | −5.7 | 14.3 | −16.0 | −0.1 | −2.5 | −0.2 | −6.8 | −14.6 |
| Other Asia, nes | 0.00654 | 6.3 | −1.4 | −4.4 | −3.8 | −17.5 | −1.8 | −0.7 | −2.1 | −16.0 | 21.1 | 0.0 |
| Spain | 0.00637 | −15.8 | 31.9 | −17.6 | −5.7 | 10.4 | 0.9 | 4.3 | −9.1 | −4.1 | −4.6 | −2.9 |
| Denmark | 0.00619 | −7.7 | 1.9 | −14.3 | −2.2 | 21.4 | −14.1 | −0.1 | −7.2 | 2.0 | −1.9 | −32.3 |
| India | 0.00589 | 14.3 | 2.1 | −5.8 | 12.2 | −8.4 | −6.8 | −14.3 | −12.6 | −6.7 | 14.9 | 0.0 |
| Korea, Rep. | 0.00547 | 8.9 | 8.0 | −2.2 | 8.4 | −16.9 | 11.3 | −17.7 | 17.2 | −19.3 | 9.1 | 0.0 |
| South Africa | 0.00506 | −6.4 | 25.2 | −6.0 | −17.8 | 0.5 | 11.7 | −20.2 | −7.2 | 22.9 | −11.7 | −1.1 |

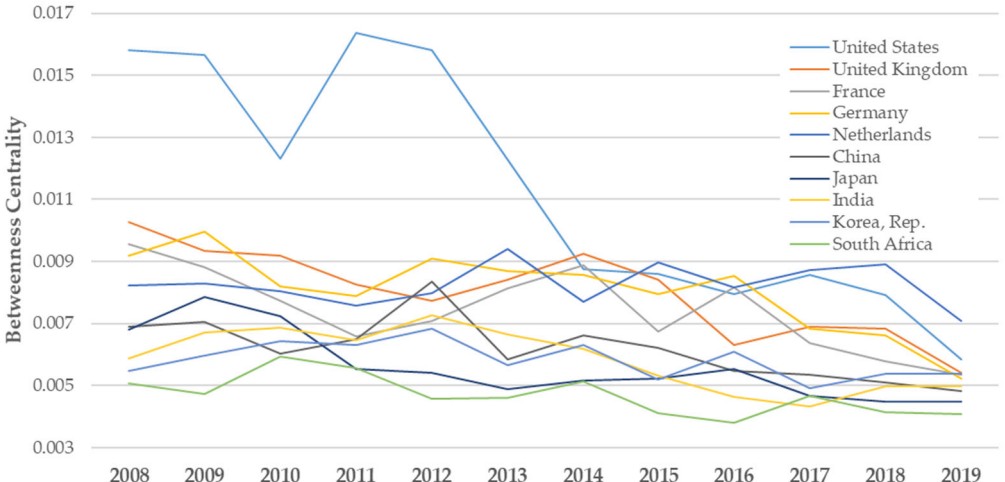

**Figure 15.** The fluctuations in betweenness centrality for 10 important countries in the world economy with highest values.

We observe that China, Japan, and South Africa are not present in Table 6 because there is a relative increase in betweenness compared to closeness. On the other hand, Malaysia, Sweden, and Thailand are present in Table 6 but are not present in Table 8 because there is a relative decrease in betweenness compared to closeness.

As betweenness centrality takes small values (0.004–0.016), we remark that there are no significant liaisons or facilitators in the studied networks. The decrease in the annual average betweenness centrality from 2008 to 2019 (Table 8) is due to the high decrease in betweenness of certain countries (the USA, the U.K., and France). This means their role as intermediaries in economic transactions became less dominant. More specifically, the Betweenness of the USA decreased by 63.1%, that of the U.K. by 47.4%, and that of France 44.2%. The largest relative decrease in betweenness was in 2014 for the USA, in 2015 for France, and in 2016 for the U.K. The smallest relative decrease in betweenness was observed for the Netherlands, Korea Rep., S. Africa, and India.

*5.6. Eigenvector Centrality*

The annual eigenvector centralities as well as the averages are presented in Table 10. The annual relative increase in eigenvector centrality is presented in Table 11, and the fluctuations of eigenvector centrality for ten selected countries are presented in Figure 16.

**Table 10.** Annual eigenvector centrality of the 18 countries with highest values. The five countries with highest centrality are indicated in light red.

| Country/Year | 2008 | 2009 | 2010 | 2011 | 2012 | 2013 | 2014 | 2015 | 2016 | 2017 | 2018 | 2019 | AVG |
|---|---|---|---|---|---|---|---|---|---|---|---|---|---|
| United States | 1.00000 | 1.00000 | 1.00000 | 1.00000 | 1.00000 | 1.00000 | 0.99689 | 1.00000 | 0.99523 | 0.99766 | 0.99593 | 0.99756 | 0.99861 |
| France | 0.99065 | 0.98588 | 0.99233 | 0.98269 | 0.98311 | 0.98738 | 1.00000 | 0.98754 | 1.00000 | 0.98804 | 0.99525 | 0.99171 | 0.99038 |
| U.K. | 0.98867 | 0.98247 | 0.99977 | 0.98497 | 0.98815 | 0.98954 | 0.99284 | 0.98423 | 0.97178 | 0.99022 | 0.99313 | 0.98769 | 0.98779 |
| Germany | 0.98372 | 0.99137 | 0.98467 | 0.98716 | 0.98948 | 0.98944 | 0.99393 | 0.98746 | 0.98990 | 0.99545 | 0.99525 | 0.98771 | 0.98963 |
| China | 0.97913 | 0.98325 | 0.98631 | 0.97124 | 0.98760 | 0.98671 | 0.99757 | 0.99642 | 0.99388 | 1.00000 | 0.99300 | 1.00000 | 0.98959 |
| Netherlands | 0.97710 | 0.97825 | 0.99176 | 0.98863 | 0.98356 | 0.99045 | 0.98639 | 0.99731 | 0.98350 | 0.99838 | 1.00000 | 0.99756 | 0.98941 |
| Japan | 0.97188 | 0.98379 | 0.98415 | 0.97432 | 0.95287 | 0.97048 | 0.95951 | 0.97470 | 0.97691 | 0.97082 | 0.98344 | 0.97133 | 0.97285 |
| Italy | 0.96358 | 0.97594 | 0.97718 | 0.98110 | 0.97343 | 0.98715 | 0.97746 | 0.98481 | 0.98490 | 0.98988 | 0.99525 | 0.98098 | 0.98097 |
| Canada | 0.96182 | 0.97980 | 0.98402 | 0.97325 | 0.96914 | 0.96002 | 0.97054 | 0.98026 | 0.96934 | 0.97533 | 0.98158 | 0.98582 | 0.97424 |
| Belgium | 0.96145 | 0.97186 | 0.95526 | 0.96102 | 0.95938 | 0.97286 | 0.96965 | 0.96628 | 0.97066 | 0.97360 | 0.97328 | 0.97990 | 0.96793 |
| Switzerland | 0.95578 | 0.93797 | 0.96029 | 0.96407 | 0.96904 | 0.95782 | 0.95564 | 0.96381 | 0.97106 | 0.97592 | 0.98030 | 0.96956 | 0.96344 |
| Spain | 0.95557 | 0.94906 | 0.98379 | 0.96885 | 0.95099 | 0.97379 | 0.96505 | 0.97752 | 0.96680 | 0.97311 | 0.99075 | 0.98390 | 0.96993 |
| Denmark | 0.94690 | 0.91480 | 0.92710 | 0.93289 | 0.91588 | 0.93845 | 0.92406 | 0.94516 | 0.94567 | 0.96555 | 0.95429 | 0.94891 | 0.93830 |
| United Arab Emirates | 0.94073 | 0.92599 | 0.96254 | 0.95449 | 0.94931 | 0.96337 | 0.94988 | 0.95981 | 0.96291 | 0.96371 | 0.97796 | 0.96356 | 0.95619 |
| India | 0.93815 | 0.95717 | 0.96292 | 0.97335 | 0.96991 | 0.96875 | 0.96421 | 0.97485 | 0.96882 | 0.97517 | 0.98790 | 0.97869 | 0.96832 |
| Australia | 0.93635 | 0.92057 | 0.93253 | 0.95625 | 0.93962 | 0.93073 | 0.96150 | 0.96115 | 0.96177 | 0.96544 | 0.96930 | 0.95825 | 0.94945 |
| Hong Kong, China | 0.93496 | 0.93749 | 0.97350 | 0.96831 | 0.96517 | 0.94851 | 0.96726 | 0.97478 | 0.97145 | 0.97487 | 0.98515 | 0.95801 | 0.96329 |
| Sweden | 0.91578 | 0.93044 | 0.93968 | 0.94862 | 0.92454 | 0.93127 | 0.94115 | 0.95053 | 0.93260 | 0.94051 | 0.95040 | 0.91965 | 0.93543 |
| AVG | 0.96123 | 0.96145 | 0.97210 | 0.97062 | 0.96506 | 0.96926 | 0.97075 | 0.97592 | 0.97318 | 0.97854 | 0.98345 | 0.97560 | |

We observed that although USA, France, the U.K., and Germany remain in the first five places with respect to the eigenvector centrality, China takes the 5th place (Table 10). The emergence of China in the first five places appears only in the eigenvector centrality, as China was not present in closeness and appeared only once in betweenness centrality, taking the 3rd place in 2012 (Table 8). The most relative decrease in the eigenvector centrality was observed in the years 2012, 2014, and 2019 (Table 11).

**Table 11.** Annual relative increase in eigenvector centrality for 18 selected countries. Each column gives the relative increase with respect to the previous year. Negative values indicate decrease.

| Country/Year | 2008 (CE) | 2009 (%) | 2010 (%) | 2011 (%) | 2012 (%) | 2013 (%) | 2014 (%) | 2015 (%) | 2016 (%) | 2017 (%) | 2018 (%) | 2019 (%) |
|---|---|---|---|---|---|---|---|---|---|---|---|---|
| United States | 1.00000 | 0.0 | 0.0 | 0.0 | 0.0 | 0.0 | −0.3 | 0.3 | −0.5 | 0.2 | −0.2 | 0.2 |
| France | 0.99065 | −0.5 | 0.7 | −1.0 | 0.0 | 0.4 | 1.3 | −1.2 | 1.3 | −1.2 | 0.7 | −0.4 |
| U. K. | 0.98867 | −0.6 | 1.8 | −1.5 | 0.3 | 0.1 | 0.3 | −0.9 | −1.3 | 1.9 | 0.3 | −0.5 |

**Table 11.** *Cont.*

| Country/Year | 2008 (CE) | 2009 (%) | 2010 (%) | 2011 (%) | 2012 (%) | 2013 (%) | 2014 (%) | 2015 (%) | 2016 (%) | 2017 (%) | 2018 (%) | 2019 (%) |
|---|---|---|---|---|---|---|---|---|---|---|---|---|
| Germany | 0.98372 | 0.8 | −0.7 | 0.3 | 0.2 | 0.0 | 0.5 | −0.7 | 0.2 | 0.6 | 0.0 | −0.8 |
| China | 0.97913 | 0.4 | 0.3 | −1.5 | 1.7 | −0.1 | 1.1 | −0.1 | −0.3 | 0.6 | −0.7 | 0.7 |
| Netherlands | 0.97710 | 0.1 | 1.4 | −0.3 | −0.5 | 0.7 | −0.4 | 1.1 | −1.4 | 1.5 | 0.2 | −0.2 |
| Japan | 0.97188 | 1.2 | 0.0 | −1.0 | −2.2 | 1.8 | −1.1 | 1.6 | 0.2 | −0.6 | 1.3 | −1.2 |
| Italy | 0.96358 | 1.3 | 0.1 | 0.4 | −0.8 | 1.4 | −1.0 | 0.8 | 0.0 | 0.5 | 0.5 | −1.4 |
| Canada | 0.96182 | 1.9 | 0.4 | −1.1 | −0.4 | −0.9 | 1.1 | 1.0 | −1.1 | 0.6 | 0.6 | 0.4 |
| Belgium | 0.96145 | 1.1 | −1.7 | 0.6 | −0.2 | 1.4 | −0.3 | −0.3 | 0.5 | 0.3 | 0.0 | 0.7 |
| Switzerland | 0.95578 | −1.9 | 2.4 | 0.4 | 0.5 | −1.2 | −0.2 | 0.9 | 0.8 | 0.5 | 0.4 | −1.1 |
| Spain | 0.95557 | −0.7 | 3.7 | −1.5 | −1.8 | 2.4 | −0.9 | 1.3 | −1.1 | 0.7 | 1.8 | −0.7 |
| Denmark | 0.94690 | −3.4 | 1.3 | 0.6 | −1.8 | 2.5 | −1.5 | 2.3 | 0.1 | 2.1 | −1.2 | −0.6 |
| United Arab Emirates | 0.94073 | −1.6 | 3.9 | −0.8 | −0.5 | 1.5 | −1.4 | 1.0 | 0.3 | 0.1 | 1.5 | −1.5 |
| India | 0.93815 | 2.0 | 0.6 | 1.1 | −0.4 | −0.1 | −0.5 | 1.1 | −0.6 | 0.7 | 1.3 | −0.9 |
| Australia | 0.93635 | −1.7 | 1.3 | 2.5 | −1.7 | −0.9 | 3.3 | 0.0 | 0.1 | 0.4 | 0.4 | −1.1 |
| Hong Kong, China | 0.93496 | 0.3 | 3.8 | −0.5 | −0.3 | −1.7 | 2.0 | 0.8 | −0.3 | 0.4 | 1.1 | −2.8 |
| Sweden | 0.91578 | 1.6 | 1.0 | 1.0 | −2.5 | 0.7 | 1.1 | 1.0 | −1.9 | 0.8 | 1.1 | −3.2 |

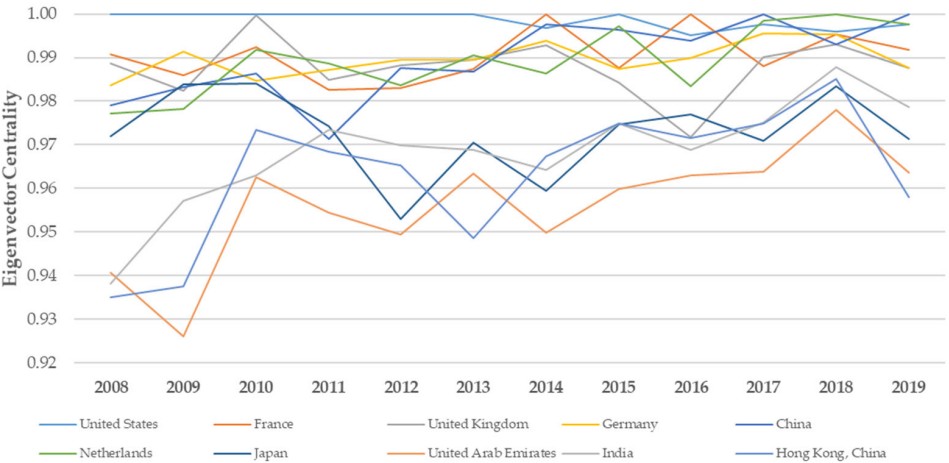

**Figure 16.** The fluctuations in eigenvector centrality for 10 important countries in the world economy with highest values.

## 6. Research Questions

The research questions Q1–Q4 (Section 2) are answered as follows:

Q1. *Network characterization of the importance of countries in the global economy.*

The world distribution of products is carried out in 239 destinations, which comprise the nodes of the network (Section 3). Their total number varied very little (<1%) from 2008 to 2019. These nodes interconnected with a total sum of edges, which exceeds 23,000 most of the time. A fully interconnected network with the same number of nodes (N = 239) would normally have 56,882 edges. The diameter and the eccentricity (Section 3.5) of the network take value three (Table 5), indicating small trade pathways, i.e., high interconnectedness.

The USA, the U.K., France, the Netherlands, China, and Germany function as mediators indirectly connecting every country in the planet to any other country (Section 3.7, Table 8 and Section 3.5, Table 5). China appeared as a mediator later.

Q2. *Collaborations between countries and global geopolitical stability as network properties and groups of countries with "strong" links.*

The search for groups of countries involved in the world trade with the modularity method (Section 3.4) shows three main groups with more than 40 member countries each and with constant presence for the specific time period and three minor communities with a small number of member countries and sporadic appearance (Figures 8 and 11). The three main groups are responsible for the distribution of the largest part of products on a global scale in such a way that the most remote destinations are three steps away from each other.

The modularity analysis of the unweighted WTN graph puts the key players (the USA, China, and Germany) in the same group as it takes into account the presence of links

instead of the size of exports. In contrast, the weighted WTN analysis takes into account the size of transactions and is closer to the real state of the world trade (Figures 9 and 10). Comparing the two different methods of network analysis, with and without weights, we observed big differences that we did not expect since we did not find anything similar anywhere in the literature. These differences were found in the number of groups that appear and in the countries from which they are composed.

The number of member countries in every group seems more dependent on basic economic indicators such as the currency rate, the GDP, and the exports of the leader country in this particular group. It is important to note that, for the period 2008–2019, the aggregate exports of key players was about 30% of the total world exports, and they had similar levels of exports.

Q3: *Changes in the global economy as changes in network properties.*

Density (Section 3.1) appeared with remarkable stability until 2018 (Figure 3, black line), indicating that the world economic network is robust (Table 2). Observing in Figure 3 with the red line the slow increase in the average clustering coefficient (Section 3.3), we conclude that there is a tendency to create connections between partner countries. The small decrease (Figure 13) in the average path length (Section 3.5) means that the distances/steps between the network nodes are slowly decreasing, indicating progressive interconnectedness that is converging towards greater globalization of the economy. In this process, the USA, the U.K., and France gradually lose their leadership, while the Netherlands and Spain remain more or less stable (Tables 6, 8 and 10).

In contrast to the closeness, betweenness changes faster and more intensely (Tables 7 and 9) since various politics can influence the route of goods and the intermediate stations. One example is the duties or restrictions that are imposed by countries on imports (2014–2016, the USA—taxes on all products coming from China) or on exports (2014–2015, Russia—embargo of products due to the occupation of Crimea).

All countries have very high eigenvector centrality (Table 10), indicating that they act as strong regulators in the international trade. Moreover, as the annual average eigenvector centrality remains more or less constant for the whole period, there are no significant changes in the regulation of the international trade.

One of the objectives of this research was to seek whether the specific analysis can capture emerging economies during the researched period. As previous studies show, using unweighted centrality measures, the increasing importance of regional trade and of some emergent countries becomes evident. Instead, with weighted centrality measures, a more traditional core–periphery picture is confirmed [15].

Q4: *Geopolitical implications of the network analysis of the WTN from 2008 to 2019.*

For the geopolitical implications of the network analysis of the WTN from 2008 to 2019, two key trends are observed:

(i) The trend towards increasing globalization of the world economy: This follows from the observed stability of average degree (Section 5.1, Table 2, and Figure 4), the slowly increasing density (Section 5.1, Table 2, and Figure 3), the increase in average clustering coefficient (Section 5.1, Table 2, and Figure 3), and the decrease in the average path length (Section 5.3, Table 5, and Figure 13);

(ii) The shift from a unipolar (USA, Tables 6–11) to a bipolar (USA–China) and even multipolar (USA–EU–BRICS) model of geopolitical power (Figures 9, 10, 14 and 15): The hegemonic position of the USA in the world economy after the fall of the Eastern block was perturbed for the following reasons:

- The emergence of a very strong competitive group with China as the dominant country (Section 5.2 and Figures 9 and 10);
- The stability and economic independence of the group of EU countries (Section 5.2 and Figure 8);
- The sporadic appearance of other groups with strong economic powers (India/South Africa, and Russia; Section 5.2 and Figures 8 and 11).

These trends have also been confirmed by other researchers using different methods of analysis [31–35].

The stable economic model of a superpower (USA) is moving into a reconstruction phase, a state in which economic instability can occur [33]. Global economic instability is very likely to have far-reaching consequences and potentially contribute to the outbreak of wars [36,37]. In the case of instability in global trading, uncertainty is increasing, the risk of investment is increasing, and war is very likely to break out [38]. Critical reasons that contribute to the conflicts between countries or nations are the following:

a. Trade wars and protectionism: In early 2018, the USA government applied and expanded tariffs on Chinese goods in response to Beijing's unfair practices, and China has retaliated, raising tariffs on U.S. exports [39,40]. It is obvious that in international geopolitics, countries raise economic borders to protect their domestic industries, which is equivalent with economic war, creating tension between nations, which is likely to trigger real war [41,42];

b. Emergence of economic alliances and blocks: In this case, due to economic instability and uncertainty, some countries may seek to form alliances and blocks in order to protect their economic interests and gain new leverage in global affairs. These blocks may cause rivalries that increase the likelihood of conflicts between opposing alliances [43,44];

c. Economic sanctions are a source of economic instability, serving as a tool of foreign policy. When countries impose severe economic sanctions, tensions increase, and military responses or escalations are provoked [45].

A typical example of a trade war that causes geopolitical instability is the case mentioned above between the USA and China. The fact that both countries have taken measures during the researched period, affecting global trade and all the other countries, emerged within the present research (Section 5.2 and Figure 8).

In Table 12, we present the calculations of our research in comparison with the previous works on the same topic as mentioned in Section 2.

**Table 12.** Comparison of our calculations with previous research on the world trade network.

| | Topology of the WTW [8] | The ITN [10] | The Evolution of the WTW [11] | Network Analysis of WT [15] | Network Analysis of IEP [16] | This Paper |
|---|---|---|---|---|---|---|
| Years/period | 2000 | 1948–2000 | 1981–2000 | 1995–2010 | 2019–2020 | 2008–2019 |
| Nodes | 179 | 76–187 | 159 | 178 | 50 | 238 |
| Avg degree | 43 | | | | | 97 |
| Links/edges | | 1494–10,252 | | 22,000 | | 20,045–23,492 |
| Density | | 0.524–0.590 | 0.55–0.65 | 0.53–0.7 | | 0.355–0.416 |
| Avg path length | 1.8 | | | | | 1.359–1.442 |
| Clust. coeff. | 0.65 | | 0.82 | | | 0.733–0.753 |
| Modularity | | | | | | 0.350–0.383 |
| Communities | | | | | 4–5 | 4–5 |

The results in the present work are very close to other works presented in Table 12. The addition of more countries/nodes to our research and of course the changes in global economic data in the following years and up to 2008 cause some small differences, especially in the average path length and clustering coefficient indicators. We also confirmed that globalization is progressing, and the trade system has become a self-organized complex system that must be considered from now on as a whole [8]. In addition, our research highlights the grouping of closely cooperating countries, and the only research related to a later period than ours found a similar structure in the WTN, with four or five communities and the same dominant countries [16].

The theoretical contribution of our work is the demonstration that network theory can identify the roles of countries in the WTN as well as the emerging groups of countries using the appropriate network indices (Section 3). Researchers may select the appropriate indices in order to obtain quantitative assessments addressing questions of interest. The results can be computed if we know the exports between countries.

Practical implications of our work include the following:

1. Stability of average degree (Conclusions, Q1);
2. Slowly increasing density (Conclusions, Q3);
3. Decreasing average path length (Conclusions, Q3).
4. The increasing clustering coefficient (Conclusions, Q3);
5. The positive value of modularity, indicating the presence of communities (Conclusions, Q2);
6. The number of communities (Conclusions, Q2);
7. The distinction between the weighted WTN and the associated unweighted graph (Section 5.1 and Figures 7, 9 and 10);
8. The remarkable qualitative similarity of the evolution of world GDP (Figure 5), with the average weighted degree (Figure 4, red line);
9. The number of member countries in the group of USA and the number of member countries in the group of China evolve in opposite ways (Figure 8a);
10. The specific countries that are members of the communities (Figures 9 and 10).

The results 1–7 confirm quantitively known assessments, while 8, 9, and 10 are novelties.

The limitation of our methodology is that we considered the global value of exports from one country to another. In order to have more accurate geopolitical implications, we intend to explore different kinds of exports and their geopolitical significance.

## 7. Conclusions and Future Work

This study encompasses a comprehensive analysis of the global economy using complex network methods, with a focus on identifying changes in countries' positions within the world trade network and interpreting these changes through geopolitical terms. As evidenced by research analysis, its contributions lie in providing novel insights into the long-term trends, detailed explanations about the natural grouping of countries, and the interplay between trade dynamics and geopolitical stability, all of which enhance our understanding of the complex global economic landscape.

Based on the current study's findings, the aim of future research is to supplement the dataset with data from 2019–2022 as well as the use of more sophisticated tools in order to enhance the understanding of the global economy using more complex network methods and geopolitical interpretations. Here are some future directions:

Long-term impact analysis: With access to data from 2019–2022, we can analyze the long-term impact of major events such as BREXIT, the COVID-19 pandemic, the different policies in the USA, and the war in Ukraine on the global economy. By tracking changes in countries' positions within the world trade network before, during, and after these events, it is possible to identify any lasting effects and trends.

Geopolitical risk assessment: Utilizing more complex network methods, we can develop models to assess geopolitical risks in the global economy. By considering changes in countries' centrality, connectivity, and trade dependencies in the world trade network, it may be possible to identify regions or countries that are more vulnerable to geopolitical shocks and economic disruptions.

Network resilience analysis: We can investigate the resilience of the world trade network in the face of various shocks and events. By applying entropy-based indicators and weighted centralities, the study can assess how the network adapts and reorganizes itself in response to geopolitical and economic changes, providing valuable insights into its stability and vulnerability.

Trade policy implications: The study can be extended to explore the implications of trade policies on the global economy. By analyzing the network dynamics and changes in countries' trade positions, we can gain a deeper understanding of the impact of trade agreements, tariffs, and other policies on international trade patterns and economic growth.

Sector-specific analysis: Focusing on specific economic sectors can provide a more granular understanding of the world trade network. We can investigate how geopolitical events and policies impact particular industries and how their positions in the network

evolve over time. This can shed light on the sector-specific vulnerabilities and opportunities in the global economy and supply chain.

Comparison with previous economic crises: To gain a broader perspective, we can compare the findings from the 2019–2022 period with previous economic crises, such as the 2008 financial crisis or other major historical events. This comparative analysis can highlight common patterns and differences, providing valuable lessons for policymakers and economists.

Predictive modeling: Using the enriched dataset and sophisticated tools, we can develop predictive models for the global economy's future trends. By incorporating historical data on geopolitical events and their impacts, these models can help forecast potential scenarios and inform decision making.

Network visualization and interactive tools: Creating interactive visualizations of the world trade network can enhance the accessibility and understanding of the research findings for policymakers, economists, and the public. These tools can be made available in the research community to engage a broader audience and facilitate further exploration of the data.

By pursuing these future research directions, this study can contribute significantly to the field of global economics, geopolitics, and complex network analysis. It can also serve as a valuable resource for policymakers and stakeholders seeking to navigate the complexities of the global economy in an ever-changing world.

**Author Contributions:** Conceptualization, G.D.P., L.M., K.D. and I.A.; methodology, G.D.P., L.M., K.D. and I.A.; software, G.D.P.; validation, G.D.P., L.M., K.D. and I.A.; formal analysis, G.D.P., L.M., K.D. and I.A.; investigation, G.D.P. and L.M.; resources, G.D.P.; data curation, G.D.P.; writing—original draft preparation, G.D.P.; writing—review and editing, G.D.P., L.M., K.D. and I.A.; visualization, G.D.P., L.M., K.D. and I.A.; supervision, L.M., K.D. and I.A.; project administration, G.D.P., L.M., K.D. and I.A.; funding acquisition. All authors have read and agreed to the published version of the manuscript.

**Funding:** This research received no external funding.

**Data Availability Statement:** The World Bank: DataBank—World Development Indicators https://databank.worldbank.org/source/world-development-indicators/preview/on.

**Conflicts of Interest:** The authors declare no conflict of interest.

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
