# Peer review of "Analyzing Global Geopolitical Stability in Terms of World Trade Network Analysis"

_information, doi:10.3390/info14080442_

Round 1

Reviewer 1 Report

Dear Authors,
My sincere congratulations on an interesting article. It is an interesting take on an often described and researched problem. However, I recommend more caution in the interpretation of the results obtained and their dependence on the method used, which is worth emphasising more in the discussion and conclusions of the article. Network and topological analyses somewhat overestimate the number of nodes and connections and underestimate the volumes connecting individual nodes and groups. Hence, there is a misconception of the hegemony of Germany, e.g. vis-à-vis the USA, and this is due to the large number of small European economies, while the cluster with the USA, which probably includes Mexico, Canada and others, e.g. Brazil, is less numerous, but the sum of flows is just as large. It, therefore, seems advisable to use network measures in parallel with those showing nominal values of trade volumes. A weakness is that the authors do not indicate the significance levels of the differences - e.g. the values in Tables 6-8 are difficult to interpret and it is not clear how significant the differences are. It is a real pity that the results obtained were not correlated with flow measures, but only once with reference to changes in GDP, which would have made it possible to verify their relevance and usefulness, at least in part; the attack remains an interesting proposal for visualising and presenting the phenomenon under study. Regarding the research questions posed - Q4, although interesting, the answer to this question does not derive directly from the results presented and research carried out, but rather expresses general views and should be based on the literature. The answers to questions Q1-Q3, although correct, should consider the issue raised earlier about the scale of the flows - they are much more variable than the topological system, which should be highlighted more in the interpretation of the results.
Best regards,

Author Response

Reviewer 1

Comment 1:

More caution in the interpretation of the results obtained and their dependence on the method used, which is worth emphasizing more in the discussion and conclusions of the article.

Answer to Comment 1:

The results obtained do not depend on the method used. This is stated clearly in the Conclusions as recommended (lines 656-673).

Comment 2:

Network and topological analyses somewhat overestimate the number of nodes and connections and underestimate the volumes connecting individual nodes and groups. Hence, there is a misconception of the hegemony of Germany, e.g. vis-à-vis the USA, and this is due to the large number of small European economies, while the cluster with the USA, which probably includes Mexico, Canada and others, e.g. Brazil, is less numerous, but the sum of flows is just as large. It, therefore, seems advisable to use network measures in parallel with those showing nominal values of trade volumes.

Answer to Comment 2:

The volume of transactions is not underestimated because the calculation of the communities of countries with modularity takes into account actual volume exports as weights. To highlight the difference with the corresponding unweighted graph, the calculation of modularity was done in parallel and added in Table 2 and in Figure 7, Figure 9 and Figure 10. Relevant comments have been added in Section 3.4 (lines 210-214), in Section 5.1 (lines 368-372), in the Conclusions (lines 618-625).

Concerning Germany’s hegemony in comparison to USA and China, we have added a remark in the Conclusions (lines 628-630).

Concerning the use of parallel network measures we added a remark in the Conclusions (lines 650-654) and a remark in Future Work (lines 739-740).

Comment 3:

A weakness is that the authors do not indicate the significance levels of the differences - e.g. the values in Tables 6-8 are difficult to interpret and it is not clear how significant the differences are.

Answer to Comment 3:

We added tables 7, 9, 11 where their columns present the relative differences (%), the Figures 14, 15, 16 and related comments (lines 532-533, 545-549, 551-554, 573-576, 578-581, 595-596).

Comment 4:

It is a real pity that the results obtained were not correlated with flow measures, but only once with reference to changes in GDP, which would have made it possible to verify their relevance and usefulness, at least in part; the attack remains an interesting proposal for visualizing and presenting the phenomenon under study.

Answer to Comment 4:

In order to correlate our results with flow measures we added the (out) flow index in Section 3.2 (lines 193-195) and we calculated it for ten important countries in the WTN. The results are presented in Section 5.1 (Figure 6) and related comments have been added in Section 5.1 (lines 342-347).

Comment 5:

Q4, although interesting, the answer to this question does not derive directly from the results presented and research carried out, but rather expresses general views and should be based on the literature.

Answer to Comment 5:

Geopolitical implications of the Network analysis of the WTN from 2008 to 2019 are presented in the Conclusions (lines 655-675).

Comment 6:

The answers to questions Q1-Q3, although correct, should consider the issue raised earlier about the scale of the flows - they are much more variable than the topological system, which should be highlighted more in the interpretation of the results.

Answer to Comment 6:

As mentioned in our response to comment 2, the answers to questions Q1-Q3 were based 1st on the network topology analysis and 2nd on the calculation of weighted measures, which take into account the scale of flows, such as Modularity and Average Weighted Degree. Moreover, as mentioned to our response to comment 4, we added the (out) flow index and we calculated it for ten important countries in the WTN.

Reviewer 2 Report

Thank you for giving me the opportunity to review this manuscript. The topic is very actual and of high interest. The design of the paper is well done and the model used is in accordance with the objectives of the research. The presentation of the results is well done and robust.

Minor recommendations:

in Conclusions  I suggest to highlight the theoretical and practical contributions of the research and the limits of the model.

Good luck!

the English language is ok in my opinion 

Author Response

Reviewer 2

Comment 1:

In Conclusions, I suggest to highlight the theoretical and practical contributions of the research.

Answer to Comment 1:

We added relevant comments in the Conclusions (lines 710-730)

Comment 2:

In Conclusions I suggest to highlight the limits of the model.

Answer to Comment 2:

We added in the Conclusions relevant comments (lines 731-733)

Reviewer 3 Report

Comments for Information-2480980

Global geopolitical stability is analyzed by testing indices about World Economic Network, which are calculated by complex network methods. The topic is interesting to readers. Main advantages and drawbacks are as follows:

Advantages:

1.     A new data set (from 2008 to 2019) is used and new indices are introduced in order to provide some updated answers to these a little old questions.

2.     Actually, quite a few new indices are introduced and answers supported by new data and these new indices are provided.

Drawbacks:

1.     Authors put forward 4 research questions Q1-Q4, but these questions look like not new ones in my view.

2.     There are seemingly no innovations for complex network method advance.

3.     Conclusions look like common senses.

Recommendations:

1.     compare your newly added indices with those indices of existing literatures

2.     compare your main conclusions with similar existing literatures

 Thus, a major revision is recommended this time.

Minor editing of English language required

Author Response

Reviewer 3

Global geopolitical stability is analyzed by testing indices about World Economic Network, which are calculated by complex network methods. The topic is interesting to readers. Main advantages and drawbacks are as follows:

Advantages:

  • A new data set (from 2008 to 2019) is used and new indices are introduced in order to provide some updated answers to these a little old questions.
  • Actually, quite a few new indices are introduced and answers supported by new data and these new indices are provided.

Drawback 1:

Authors put forward 4 research questions Q1-Q4, but these questions look like not new ones in my view.

Answer to Drawback 1:

The present work does not pose new scientific questions. We use Computational Physics Methods, particularly complex networks, to reach conclusions confirmed by scientists in other fields [31]-[35]. The theoretical and practical implications as well as the limitations of our work are presented in the Conclusions (lines 710-733).

Drawback 2:

There are seemingly no innovations for complex network method advance.

Answer to Drawback 2:

We agree, there is no innovation on mathematical network theory. We used established tools in order to provide innovative answers to questions Q1-Q4.

The theoretical and practical implications as well as the limitations of our work are presented in the Conclusions (lines 710-733).

Drawback 3:

Conclusions look like common senses.

Answer to Drawback 3:

Indeed, the implications 1-7 (lines 716-724) have been mentioned in the literature. We simply confirm all these results quantitavely using the methodology of network theory. However, the results 8, 9, 10 (lines 725-729) are novelties.

The theoretical and practical implications as well as the limitations of our work are presented in the Conclusions (lines 710-733).

Recommendation 1:

Compare your newly added indices with those indices of existing literatures.

Answer to Recommendation 1:

We added in the Conclusions Table 9 and lines 701-709.

Recommendation 2:

Compare your main conclusions with similar existing literatures.

Answer to Recommendation 2:

We added in the Conclusions Table 9 and lines 701-709. The references are also included previously (lines 71-73, 78-80, 89-90, 96-97, 128-132).

Recommendation 3:

Thus, a major revision is recommended this time.

Answer to Recommendation 3:

Done.

Recommendation 4:

Minor editing of English language required.

Answer to Recommendation 4:

Done.

Round 2

Reviewer 3 Report

Conclusions and comparisons with existing studies are strenghened. Thus, I have no more comments then. 

Language is good now.